# Efficiently Deciding Algebraic Equivalence of Bow-Free Acyclic Path Diagrams

**Thijs van Ommen**[1]

[1]Information and Computing Sciences, Utrecht University, Utrecht, The Netherlands

## Abstract

For causal discovery in the presence of latent confounders, constraints beyond conditional independences exist that can enable causal discovery algorithms to distinguish more pairs of graphs. Such constraints are not well-understood yet. In the setting of linear structural equation models without bows, we study algebraic constraints and argue that these provide the most fine-grained resolution achievable. We propose efficient algorithms that decide whether two graphs impose the same algebraic constraints, or whether the constraints imposed by one graph are a subset of those imposed by another graph.

## 1 INTRODUCTION

Causal discovery is the problem of learning a causal graph from data. This is a difficult problem for many reasons, including the danger of drawing wrong conclusions due to noisy data, the superexponential size of the search space, and the fact that some graphs are just indistinguishable based on data alone.

A further complication is that in many situations, we cannot safely assume *causal sufficiency*: the assumption that we have measurements of all variables that are relevant for explaining the statistical relations we see in the data. A *latent confounder* is a variable that is not observed, but is a cause of two or more observed variables. If we fail to take the possible existence of latent confounders into account, we would wrongly try to explain the statistical relation between the observed variables in terms of causal relations between them, when in fact there might not be such relations.

For a graph without latent variables, its statistical model can be fully described by a list of (conditional) independences that must hold between the variables. Thus, looking for such independences in the data will allow us to differentiate between any pair of graphs that we could theoretically distinguish. For types of graphs that allow latent variables, this is no longer enough, as new constraints such as the Verma constraint [Robins, 1986, Verma and Pearl, 1991] may be imposed on the statistical model. Taking such constraints into account could help us distinguish between more graphs.

In this paper, we study *algebraic* constraints arising in linear structural equation models for a class of graphs known as bow-free acyclic path diagrams. In particular, we are interested in the following question: given two bow-free graphs, are they distinguishable based on algebraic constraints? Two graphs that are indistinguishable in this way are called *algebraically equivalent* [Van Ommen and Mooij, 2017].

An algorithm that answers this question efficiently would have many applications. For example, in a score-based causal discovery search, it could be used to avoid the expensive operation of scoring a graph that is equivalent to one we have already seen. Also, when evaluating the performance of a causal discovery method on simulated data, we often face the problem that the algorithm might output a single graph as representative of an equivalence class, and to assess this output, we need to know if the output graph is algebraically equivalent to the graph from which the data were simulated. The algorithms we propose can be used for these purposes.

The rest of this paper is structured as follows. After discussing related work in Section 1.1 and preliminaries in Section 2, we will define efficient algorithms in Section 3.[1] These algorithms can decide whether a given graph imposes a given algebraic constraint; whether one graph imposes all the algebraic constraints that another one imposes; and whether two graphs are algebraically equivalent. In Section 4, we discuss other equivalence relations that could be used for causal discovery, and argue that for bow-free

---

[1]An implementation of these algorithms can be found at `https://github.com/UtrechtUniversity/aelsem_decide`.

acyclic path diagrams, algebraic equivalence might be the most appropriate. We also prove some necessary and sufficient conditions for algebraic equivalence in Section 4.1.1. Finally, Section 5 describes some small experiments, and a discussion and conclusion are in Sections 6 and 7.

## 1.1 RELATED WORK

Claassen and Bucur [2022] present an algorithm that decides Markov equivalence, i.e. the more coarse-grained notion that only takes conditional independences into account. This algorithm is very fast ($O(n)$) for sparse graphs. For general graphs, it is $O(n^4)$, which is similar to our algorithms.

For algebraic equivalence, no efficient algorithms exist yet. Nowzohour et al. [2017] test 'empirical equivalence' by computing the maximum likelihood scores of two graphs and calling them equivalent if these scores are within some tolerance. Scoring a graph is an expensive operation requiring iterative optimization algorithms even for linear structural equation models [Drton et al., 2009], and the result is not reliable due to numerical inaccuracy and because the likelihood may have spurious local maxima [Drton and Richardson, 2004]. We include an experimental comparison to this method in Section 5.

None of these methods can be used to decide whether one model contains another, in the sense that all algebraic constraints imposed by one are also imposed by the other. Our Algorithm 2 in Section 3.2 can answer this question for two bow-free acyclic path diagrams, which may be useful in its own right.

Our algorithms may also be applicable to discrete and non-parametric models. The relevant notion of equivalence in this case is *nested Markov equivalence*, a refinement of Markov equivalence. We present a partial result on this in Section 4.2.

## 2 PRELIMINARIES

Graphical models are useful for modelling the statistical relations between a set of variables, and more specifically also for modelling causal relations [Pearl, 2000]. The most basic class of graphs used for this purpose is that of *directed acyclic graphs (DAGs)*. A DAG $G$ consists of a set of nodes $V$ and a set of directed edges $E$ which do not form directed cycles $v \to \ldots \to v$. Interpreted causally, the presence of a directed path $v \to \ldots \to w$ in $G$ indicates that random variable $X_v$ is a *cause* of $X_w$: an external intervention on $X_v$ is expected to lead to a change in the distribution of $X_w$.

*Directed mixed graphs (DMGs)* have been used to model the presence of latent confounders without including them explicitly as extra variables in the model, first by Wright [1921]. These graphs have *bidirected edges* in addition to

directed ones. A bidirected edge $v \leftrightarrow w$ indicates the existence of a latent variable that is a cause of both $X_v$ and $X_w$. A DMG with no directed cycles is called an *acyclic DMG (ADMG)*. An ADMG is called a *bow-free acyclic path diagram (BAP)* if it also does not contain a *bow*, which is the co-occurrence of a directed edge $v \to w$ and a bidirected edge $v \leftrightarrow w$ between a single pair of nodes. In other words, BAPs are *simple* ADMGs, i.e. they have no multiple edges.

A *linear structural equation model (LSEM)* is a model on a set of real-valued random variables $\{X_v \mid v \in V\}$ by means of a DMG $G$, describing their joint distribution via

$$X_v = \lambda_{0v} + \sum_{w \in \mathrm{pa}_G(v)} \lambda_{wv} X_w + \epsilon_v.$$

Here, $\mathrm{pa}_G(v)$ denotes the set of parents of $v$ in the graph $G$: those vertices $w$ that have a directed edge to $v$. The $\epsilon$'s are noise terms, which have $\mathrm{Var}(\epsilon_v) = \omega_{vv}$, and for $v \neq w$ must have $\mathrm{Cov}(\epsilon_v, \epsilon_w) = 0$ unless there is a bidirected edge between $v$ and $w$; then $\mathrm{Cov}(\epsilon_v, \epsilon_w) = \omega_{vw}$. The $\lambda$'s and $\omega$'s are parameters of the model. Dropping the intercepts $\lambda_0$. because they have no influence on $\Sigma = \mathrm{Cov}(\mathbf{X})$, the parameters can be represented as matrices $\Lambda$ and $\Omega$, which may have nonzero entries only in the following places: $\Lambda_{vw}$ is allowed to be nonzero if there is a directed edge from $v$ to $w$ in $G$, and $\Omega_{vw}$ can be nonzero if $v = w$ or there is a bidirected edge between $v$ and $w$. Being a covariance matrix, $\Omega$ must be symmetric and positive definite. We will only consider graphs without directed cycles in this paper; for such graphs, $(I - \Lambda)$ is always invertible.

The noise terms $\epsilon_v$ are often assumed to be Gaussian, but this assumption is not necessary for the theory developed in this paper because we will look at the data only through the covariance matrix $\Sigma$. This does mean that if the data is not Gaussian, we ignore information present in higher-order moments. This information is potentially valuable: Wang and Drton [2023] show that if the distributions are sufficiently non-Gaussian, all BAPs can be distinguished from each other using higher-order moments. These moments can be captured in tensors and analyzed algebraically; see e.g. [Améndola et al., 2023].

For parameters $\Lambda, \Omega$, we can compute $\Sigma = \mathrm{Cov}(\mathbf{X})$ as

$$\Sigma = \phi(\Lambda, \Omega) = (I - \Lambda)^{-T} \Omega (I - \Lambda)^{-1}, \qquad (1)$$

where $^{-T}$ denotes the transposed inverse; see e.g. [Foygel et al., 2012]. Now we can define the *model* $\mathcal{M}(G)$ of a graph as

$$\mathcal{M}(G) = \{\phi(\Lambda, \Omega) \mid \Lambda \text{ and } \Omega \text{ compatible with } G\}.$$

The parameterization map $\phi$ can also be understood graphically using the concept of a *trek*, which is a path without colliders (i.e. two consecutive edges along a trek do not both have an arrowhead into the node between them on the path).

Equivalently, a trek consists of any number of directed edges traversed in the backward direction, then optionally a bidirected edge, then any number of directed edges traversed in the forward direction. The *trek rule* is

$$\sigma_{vw} = \sum_{\substack{\text{treks } \tau \\ \text{between } v \text{ and } w}} \Big( \prod_{x \leftarrow y \in \tau} \lambda_{yx} \cdot \omega_\tau \cdot \prod_{x \rightarrow y \in \tau} \lambda_{xy} \Big), \quad (2)$$

where $\omega_\tau = \omega_{xy}$ if $x \leftrightarrow y \in \tau$; otherwise $\omega_\tau = \omega_{cc}$ where $c$ is the unique node in $\tau$ with no incoming edges.

Similar to treks, a *half-trek* from $v$ to $w$ is either a directed path from $v$ to $w$, or a bidirected edge $v \leftrightarrow x$ followed by a directed path from $x$ to $w$. We write $w \in \text{htr}(v)$ if $w$ is reachable by a half-trek from $v$. The *half-trek criterion (HTC)* of Foygel et al. [2012] will play a role in our theory. A graph satisfying this criterion is called *HTC-identifiable*. All BAPs are HTC-identifiable; many ADMGs and some DMGs are HTC-identifiable as well. Foygel et al. present an algorithm that, given an HTC-identifiable graph $G$ and a $\Sigma \in \mathcal{M}(G)$, will almost always find parameters $\Lambda$ and $\Omega$ for $G$ such that $\Sigma = \phi(\Lambda, \Omega)$.

We are motivated by the problem of *causal discovery*: we want to use data sampled from $\mathbf{X}$ to learn which graph is behind the data-generating process. In practice, we often are unable to distinguish between several graphs that can explain the data equally well because they are *distributionally equivalent*: $\mathcal{M}(G) = \mathcal{M}(G')$.

As $\Sigma$ is defined by polynomials, also $\mathcal{M}(G)$ can be described as the set of all positive definite $\Sigma$ that satisfy some polynomial equalities ($f_i(\Sigma) = 0$) and inequalities ($g_i(\Sigma) > 0$) (or $\mathcal{M}(G)$ may be the union of finitely many such sets). Such objects are studied in algebraic geometry [Cox et al., 2015]. A useful simplification is to drop all inequality constraints, thus allowing some $\Sigma$ that were not in $\mathcal{M}(G)$. The result is called the *algebraic model* and written $\overline{\mathcal{M}}(G)$. We will see in Section 4.1 that for BAPs, the difference between $\mathcal{M}(G)$ and $\overline{\mathcal{M}}(G)$ is very small. The retained polynomial equalities are also called *algebraic constraints*. If a model satisfies algebraic constraints $f_1$ and $f_2$, we see it also satisfies $f_1 + f_2$ and $g \cdot f_1$, where $g$ can be any polynomial. A set of polynomials that is closed under these operations is called an *ideal*, and the smallest ideal containing some set of polynomials $f_1, \ldots, f_k$ is said to be *generated by* that set. Two graphs $G$ and $G'$ are called *algebraically equivalent* if $\overline{\mathcal{M}}(G) = \overline{\mathcal{M}}(G')$ [Van Ommen and Mooij, 2017].

We list some examples of algebraic constraints to illustrate their generality:

**Vanishing correlation** The polynomial is simply $\sigma_{vw}$. For multivariate Gaussians, $\sigma_{vw} = 0$ is equivalent to marginal independence.

**Vanishing partial correlation** The partial correlation $\rho_{vw \cdot S}$ between $v$ and $w$ controlling for $S$ is zero iff

the numerator $|\Sigma_{\{v\} \cup S, \{w\} \cup S}|$ in its definition is zero. This determinant is a polynomial in $\Sigma$. For multivariate Gaussians, this polynomial vanishes iff $v$ and $w$ are conditionally independent given $S$.

**Vanishing minor constraints** Generalizing the above, Sullivant et al. [2010] consider constraints of the form $|\Sigma_{A,B}|$ for arbitrary minors of $\Sigma$, and give a graphical characterization for such constraints in terms of t-separation, which generalizes the well-known d-separation.

**Graphically representable constraints** Van Ommen and Drton [2022] show that many constraints arising in LSEMs can be expressed as determinants of matrices constructed from $\Sigma$, with each entry in this matrix being either $\sigma_{vw}$ or 0. These matrices may be larger than $n \times n$, the size of $\Sigma$. The zero/nonzero pattern of the matrix can be thought of as the adjacency matrix of a bipartite graph. These 'graphical representations' give these constraints their name.

## 2.1 THE GRAPHICALLY REPRESENTED IDEAL

For a given graph $G$, we would like to have a set of algebraic constraints that together generate the ideal of $\overline{\mathcal{M}}(G)$. This task can be done by methods from algebraic geometry [Cox et al., 2015], but these are very slow, possibly taking hours even for graphs with 4 or 5 nodes. Van Ommen and Drton [2022] outline a procedure that, given an HTC-identifiable graph, outputs a list of graphical representations of constraints. For a BAP with $n$ vertices and $m$ edges, this is a list of $\binom{n}{2} - m$ constraints, i.e. one per pair of nonadjacent nodes. We will call the ideal generated by these constraints the *graphically represented ideal*. These ideals do not always describe the algebraic model perfectly: they may have *spurious components* which allow the existence of sets of $\Sigma$'s that satisfy the graphically represented constraints, yet are not in the algebraic model. If no such spurious $\Sigma$'s are positive definite, the ideal is called *PD-primary*; if the spurious $\Sigma$'s do not include the identity matrix, the ideal is called *I-primary*. For general graphs, the graphically represented ideal may fail to be PD- or *I*-primary. We illustrate this by Examples 1 and 2 below, where we see spurious $\Sigma$'s for two graphs. Additional discussion of these examples can be found in Appendix A.

**Example 1.** The graph in Figure 1(a) is a BAP and its graphically represented ideal is *I*-primary. It is not PD-primary: the ideal permits

$$\Sigma = \begin{bmatrix} 1 & 3/4 & 2/9 & 0 & 0 \\ 3/4 & 1 & 3/4 & 0 & 0 \\ 2/9 & 3/4 & 1 & 0 & 0 \\ 0 & 0 & 0 & 1 & 1/2 \\ 0 & 0 & 0 & 1/2 & 1 \end{bmatrix},$$

which is positive definite but clearly not in the model, as it has $\sigma_{de} \neq 0$ while node $e$ is isolated.

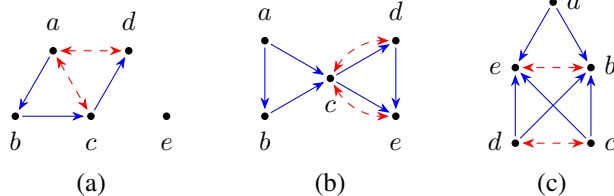

Figure 1: (a) A BAP for which the graphically represented ideal is $I$-primary but not PD-primary; (b) an ADMG for which the graphically represented ideal is not $I$-primary; (c) a BAP whose model may be mistakenly classified as a submodel of (b)'s model due to the latter's spurious components.

**Example 2.** Consider the graph in Figure 1(b). This graph is not a BAP, but is HTC-identifiable so that a graphically represented ideal can be found. In this case, such an ideal will be neither PD- nor $I$-primary. The set of points that satisfy the graphically represented constraints contains the set $\{\Sigma \mid \sigma_{ac} = \sigma_{ad} = 0\}$, even though most $\Sigma$'s in this set are not actually in the model and are thus spurious. Note that this set is precisely the model of the graph in Figure 1(c). So we see that in this case, the set of points that satisfy the graphically represented constraints is now so much larger than the model that it contains another model; in fact, one of the same dimensionality.

Van Ommen and Drton [2022] show that for ancestral graphs [Richardson and Spirtes, 2002], the graphically represented ideal is PD-primary, and for BAPs, it is $I$-primary.

## 2.2 $I$-PRIMARY IDEALS ENABLE MODEL INCLUSION TESTING

We see in Example 2 that the spurious component of a non-$I$-primary ideal for model $\overline{\mathcal{M}}(G')$ may allow a set of $\Sigma$'s large enough to contain another model $\overline{\mathcal{M}}(G)$ in its entirety. This would pose a problem for our algorithms: to decide whether $\overline{\mathcal{M}}(G) \subseteq \overline{\mathcal{M}}(G')$, we want to detect if there is a point in $\overline{\mathcal{M}}(G) \setminus \overline{\mathcal{M}}(G')$, but all such points might be 'hidden' behind a spurious component. As the following theorem shows, for $I$-primary ideals, $\overline{\mathcal{M}}(G) \setminus \overline{\mathcal{M}}(G')$ cannot be completely covered by a spurious component in this way. More strongly, generic points in $\mathcal{M}(G)$ will not be covered by spurious components.

**Theorem 1.** *Let $J$ be an $I$-primary ideal for $\overline{\mathcal{M}}(G')$. Let $\mathcal{M}(G)$ be another graphical model. Then $\mathcal{M}(G) \cap V(J) \setminus \overline{\mathcal{M}}(G')$ is of lower dimension than $\mathcal{M}(G)$.*

$V(J)$ denotes the set of points $\Sigma$ that are zeros of all polynomials in the ideal $J$. Note that $V(J) \setminus \overline{\mathcal{M}}(G')$ is the set of points covered by spurious components of $J$. See Cox et al. [2015] for the definition of dimension in this context. The proof of this theorem is provided in Appendix B.

Our algorithms are built on this, and on the fact that graphically represented ideals of BAPs are $I$-primary.

## 3 ALGORITHMS

In this section, we introduce three algorithms and prove their correctness and efficiency. Algorithm 1 decides whether a graph imposes a specified algebraic constraint. Algorithm 2 compares two graphs, and decides whether the algebraic model of the first is contained in that of the second. Finally, Algorithm 3 decides whether two graphs are algebraically equivalent.

The algorithms we will introduce are randomized algorithms. Specifically, they are Monte Carlo algorithms with one-sided error: when given an input for which the correct answer is 'true', they will always correctly answer 'true', but when given an input for which the correct answer is 'false', there is a small probability that they incorrectly output 'true' (i.e. a false positive).

The probability $q$ of an incorrect answer depends on the input, and for each algorithm we prove an upper bound on this probability in the theorems below. If a higher degree of confidence is desired, the algorithm can be run repeatedly, sampling new, independent random values each time, until it outputs 'false' once or 'true' $k$ times. In the former case, we can be sure of the correctness of the answer; in the latter case, the probability of error has been reduced to $q^k$.

### 3.1 TESTING A CONSTRAINT

The problem of testing whether a graph imposes a constraint can be thought of as the analogue to testing a d-separation in a DAG, generalized from DAGs to ADMGs and from (conditional) independence constraints to algebraic constraints.

Intuitively, to decide whether a graphical model $\overline{\mathcal{M}}(G)$ imposes a constraint, we can take a random point $\Sigma$ in $\mathcal{M}(G)$ by choosing random values for the model's parameters. If we find a $\Sigma$ that does not satisfy the constraint, we conclude that the model does not impose this constraint. If $\Sigma$ does satisfy the constraint, we are not sure, but using that a polynomial that is not identically zero will assume nonzero values in 'most' places, we have evidence that the constraint is zero, thus satisfied, for all $\Sigma \in \overline{\mathcal{M}}(G)$. This is the essence of Algorithm 1. The word 'most' above can be made precise in different ways: using the concept of dimension as in Theorem 1, or by bounding the number of zeros in certain finite regions. The latter is what we use in our proofs.

In order to implement this idea in an algorithm, we have to make a choice of what parameter values to sample:

- We can sample real-valued numbers (or in practice, floating-point numbers) and compute with those. This has the disadvantage that we have to be aware of nu-

merical error in the computations. As such, if we find that $f(\Sigma)$ is not exactly zero but within some tolerance, we have to return 'true', increasing the probability of error if actually the constraint is not satisfied.

- To avoid numerical issues, we can sample integer values. The computation of $\Sigma$ and then of $f(\Sigma)$ takes the form of a polynomial with possibly large degree. So if we sample from a large range of integers, the intermediate results will not fit into a computer word and arithmetic operations become slower. If we sample from a small range, again the probability of error increases.

- We can sample and compute with elements of the finite field $\mathbb{F}_p$ for a sufficiently large prime $p$, i.e. carrying out all computations modulo $p$ [von zur Gathen and Gerhard, 2013]. A suitable choice is $2^{31}-1$: this allows all arithmetic operations to be implemented efficiently on any 64-bit computer. Because we only have to return 'true' if the computation comes out as *exactly* 0 modulo $p$, the probability of error is extremely small.

Clearly, it is advantageous to work with $\mathbb{F}_p$. The algorithms in this section take $p$ as an input. Theorems 2 and 3 will make precise what values of $p$ are 'sufficiently large', and how confident we can be when we receive a 'true' output. By choosing $p$ large enough, we can ensure the probability of error is below any desired bound. For example, for the choice $p = 2^{31} - 1$ suggested above and 'small' inputs (e.g. graphs of five nodes), all algorithms have a one-sided probability of error less than $4.61 \cdot 10^{-8}$.

In $\mathbb{F}_p$, there is no distinction between positive and negative numbers. As a result, the concept of positive definiteness is not well-defined, and we do not require such a property of the 'covariance' matrices that appear in our algorithms. We could not rely on positive definiteness to begin with: for BAPs, the graphically represented ideal may fail to be PD-primary as in Example 1, meaning that among $\Sigma$ that satisfy the constraints yet are outside the algebraic model, also positive definite examples will exist.

**Theorem 2.** *Algorithm 1 has one-sided probability of error at most $(2\ell_G + 1)\deg(f)/p$, where $\ell_G$ is the length of the longest directed path in $G$ and $\deg(f)$ is the degree of $f$. For a constraint expressed as the determinant of a $\deg(f) \times \deg(f)$ matrix, it runs in time $O(n^\omega + \deg(f)^\omega)$, where $\omega$ is the matrix multiplication exponent.[2]*

---

[2]The straightforward matrix multiplication algorithm is $O(n^3)$. Asymptotically more efficient algorithms exist: Strassen's algorithm [1969] attains $\omega \approx 2.81$, and algorithms based on the one by Coppppersmith and Winograd [1990] attain $\omega \approx 2.37$. The best known lower bound is $\omega \geq 2$. However, due to the large hidden constants, these algorithms only become practically useful for large matrices. Strassen's algorithm is only viable for $n$ in the hundreds [Huang et al., 2016], and Coppersmith–Winograd-like algorithms are currently not practical at all. So for the matrices

---

**Algorithm 1:** Decide whether a graphical model satisfies a constraint.

**Input:** An ADMG $G$, an algebraic constraint $f$ (a polynomial in $\Sigma$), and a prime $p$
**Output:** If for all $\Sigma \in \mathcal{M}(G)$, $f(\Sigma) = 0$, output `true`; otherwise, with large probability output `false`

Sample $\Lambda$ and $\Omega$ for $G$ uniformly at random from $\mathbb{F}_p$;
Let $\Sigma = (I - \Lambda)^{-T}\Omega(I - \Lambda)^{-1}$;
**if** $f(\Sigma) = 0$:
    **return** `true`;    *// Evidence constraint is satisfied*
**else:**
    **return** `false`; *// Constraint definitely not satisfied*

---

*Proof.* Clearly, the first lines of the algorithm sample a $\Sigma$ from $\mathcal{M}(G) \subseteq \overline{\mathcal{M}}(G)$. We see that if $f(\Sigma) = 0$ for all $\Sigma \in \overline{\mathcal{M}}(G)$, the algorithm always outputs 'true'.

Now consider the case that $\overline{\mathcal{M}}(G)$ does not satisfy $f$. The computation performed by the algorithm is the composition of two polynomials: $g(\Lambda, \Omega) = f(\phi(\Lambda, \Omega))$. The degree of $g$ is bounded by the product of the degrees of $f$ and $\phi$. Using the trek rule (2), we can bound the degree of $\phi$ by $(2\ell_G + 1)$, which is an upper bound on the degrees of the monomials that appear there. This bounds the degree of $g$ by $(2\ell_G + 1)\deg(f)$. As $g$ is not the zero polynomial, we apply the Schwartz–Zippel lemma [Schwartz, 1980][3] to find that

$$P[g(\Lambda, \Omega) = 0 \mid g \not\equiv 0] \leq \frac{1}{p}(2\ell_G + 1)\deg(f).$$

The tasks of computing products, inverses, and determinants of $n \times n$ matrices can each be done in time $O(n^\omega)$ [Bunch and Hopcroft, 1974]. This shows that for a constraint expressed as the determinant of a $\deg(f) \times \deg(f)$ matrix, Algorithm 1 runs in time $O(n^\omega + \deg(f)^\omega)$. $\qquad\square$

## 3.2 TESTING MODEL INCLUSION

Algorithm 2 takes as input two graphs $G$ and $G'$ (of which $G'$ must be a BAP) and decides whether $\overline{\mathcal{M}}(G) \subseteq \overline{\mathcal{M}}(G')$, i.e., whether all algebraic constraints imposed by $\overline{\mathcal{M}}(G')$ are also imposed by $\overline{\mathcal{M}}(G)$. It builds on the techniques used in Algorithm 1, but also requires some new ideas.

First, we need an efficiently computable description of $\overline{\mathcal{M}}(G')$. For this purpose, we use the graphically represented ideal described by Van Ommen and Drton [2022] and discussed in Section 2.1. The graphically represented

---

considered here, in practice $\omega = 3$.

[3]The lemma is known by that name because a very similar result was shown independently by Zippel [1979], though we use the bound of Schwartz [1980] which is stronger in our case.

ideal is based on the 'rational constraints' of Van Ommen and Mooij [2017]. The intuition behind these is that for the $\Sigma$ that is sampled randomly from the model of $G$, we will try to find parameters $\Lambda', \Omega'$ for $G'$ that would establish that $\Sigma \in \mathcal{M}(G')$. First, $\Lambda'$ is computed using the HTC-identification algorithm of Foygel et al. [2012]. This algorithm will always assign 0's to elements of $\Lambda'$ that should be 0, i.e., those that do not correspond to directed edges in $G'$. Next, $\Omega'$ is computed as $(I - \Lambda')^T \Sigma (I - \Lambda')$. This computation does not check where in $\Omega'$ it places nonzeros. If $\Sigma \in \mathcal{M}(G')$, then $\Omega'$ will have its nonzeros only in permissible places, namely on the diagonal and in places where $G'$ has bidirected edges. But if $\Sigma \notin \mathcal{M}(G')$, $\Omega'$ will typically have nonzeros in certain other places as well. Computing the values of these other elements of $\Omega'$ amounts to evaluating each of the rational constraints. The rational constraints do not describe the model perfectly: as Example 2 demonstrates, this algorithmic approach could give the wrong answer if we did not restrict $G'$ to be bow-free.

The graphically represented constraints differ from the rational constraints in that the graphically represented constraints are polynomials in $\Sigma$, while computing $\Lambda'$ (and thus $\Omega'$) from $\Sigma$ also requires divisions. Algorithm 2 avoids these divisions by computing polynomial multiples of $\Lambda'$ and $\Omega'$ instead, thereby mimicking the computation of Van Ommen and Drton [2022] exactly. Thus rather than $\Omega'$, Algorithm 2 computes the matrix $\tilde{\Omega}'$, whose entries are multiples of $\Omega'$. Because $I - \Lambda'$ plays a more central role in this computation than $\Lambda'$, it is convenient in Algorithm 2 to work with $\tilde{\Lambda}'$, which equals $I - \Lambda'$ except that each row is multiplied by some polynomial.

Algorithm 2 further differs from Algorithm 1 in that it does not construct the constraints one by one, but evaluates them jointly as outlined above to avoid redundant computation between the constraints as well as within single constraints. This leads to a significant speedup: the graphically represented constraints can have degrees that are exponential in the number of nodes of $G'$, but with this more efficient computation, the algorithm remains polynomial-time. For this reason, also for the task of testing a constraint $f$, it may be preferable to use Algorithm 2 rather than Algorithm 1, supplying as input $G'$ a graph that imposes $f$ as its only algebraic constraint.

**Theorem 3.** *Algorithm 2 has one-sided probability of error at most*

$$\frac{1}{p}(2\ell_G + 1)\Big(1 + \max_{\substack{\{v,w\} \\ \textit{nonadjacent in } G'}} (a_v + a_w)\Big),$$

*where $\ell_G$ is the length of the longest directed path in $G$ and*

$$a_v = |\mathrm{pa}_{G'}(v)| + \sum_{w \in \mathrm{pa}_{G'}(v) \cap \mathrm{htr}_{G'}(v)} a_w$$

*if* solve($v$) *was called, and* $a_v = 0$ *otherwise. The runtime of Algorithm 2 is $O(n^{\omega+1})$.*

---

**Algorithm 2:** Decide whether one algebraic model is contained in another.

**Input:** An ADMG $G$, a BAP $G'$, and a prime $p$
**Output:** If $\overline{\mathcal{M}}(G) \subseteq \overline{\mathcal{M}}(G')$, output `true`; otherwise, with large probability output `false`

Sample $\Lambda$ and $\Omega$ for $G$ uniformly at random from $\mathbb{F}_p$;
Let $\Sigma = (I - \Lambda)^{-T} \Omega (I - \Lambda)^{-1}$;
Let $\tilde{\Lambda}' = I_n$;
**for** $v \in V$ with $\deg_{G'}(v) < n - 1$:
    solve($v$);
Let $\tilde{\Omega}' = \tilde{\Lambda}'^T \Sigma \tilde{\Lambda}'$;
**if** $\tilde{\Omega}'_{vw} = 0$ for all $\{v, w\}$ nonadjacent in $G'$:
    **return** `true`;   // *Evidence that $\overline{\mathcal{M}}(G) \subseteq \overline{\mathcal{M}}(G')$*
**else:**
    **return** `false`;      // *Definitely $\overline{\mathcal{M}}(G) \not\subseteq \overline{\mathcal{M}}(G')$*

**def** solve($v$):
    // *Compute and store the correct value for $\tilde{\Lambda}_{\cdot,v}$.*
    **if** solve($v$) was called previously:
        **return**;
    **if** $\mathrm{pa}_{G'}(v) = \varnothing$:
        **return**;
    **for** $w \in \mathrm{pa}_{G'}(v) \cap \mathrm{htr}_{G'}(v)$:
        solve($w$);
    Define matrix $\mathbf{M}^{(v)}$ with a row for each
    $w \in \mathrm{pa}_{G'}(v)$ and $n$ columns by
$$\mathbf{M}_{w,\cdot}^{(v)} = \begin{cases} \tilde{\Lambda}'_{\cdot,w} & \text{if } w \in \mathrm{htr}_{G'}(v) \\ I_{\cdot,w} & \text{otherwise} \end{cases};$$
    Let $\mathbf{A}^{(v)} = \mathbf{M}^{(v)} \cdot \Sigma_{\cdot,\mathrm{pa}_{G'}(v)}$;
    Let $\mathbf{b}^{(v)} = \mathbf{M}^{(v)} \cdot \Sigma_{\cdot,v}$;
    Let $\tilde{\Lambda}'_{v,v} = |\mathbf{A}^{(v)}|$, and for each $w \in \mathrm{pa}_{G'}(v)$,
    $\tilde{\Lambda}'_{w,v} = -|\mathbf{A}_w^{(v)}|$ where $\mathbf{A}_w^{(v)}$ is obtained from
    $\mathbf{A}^{(v)}$ by replacing column $w$ by $\mathbf{b}^{(v)}$;

---

As the $a_v$-terms in the error bound need to be computed separately for each graph, it may be useful to have a bound that holds over all graphs, depending only on the number of vertices $n$.

**Lemma 4.** *For $n \geq 4$, the probability of error in Algorithm 2 is at most*

$$\frac{1}{p}(2n - 1)\left(\frac{3}{8}2^n - 1\right).$$

The proof of these results is given in Appendix B.

For $n = 5$, Lemma 4 gives the bound $4.61 \cdot 10^{-8}$ on the error probability (using $p = 2^{31} - 1$); with this bound, it may be acceptable to run the algorithm only once. For $n = 25$, the bound is 0.29; then the algorithm will need to be run repeatedly to reduce the probability of error, or slower arithmetic may need to be accepted to accommodate a larger

**Input:** Two BAPs $G$ and $G'$, and a prime $p$
**Output:** If $\overline{\mathcal{M}}(G) = \overline{\mathcal{M}}(G')$, output `true`; otherwise,
   with large probability output `false`

**if** $G$ and $G'$ have different skeletons**:**
   **return** `false`;         // *Definitely no equivalence*
**elif** Algorithm 2 returns `true` for $G$, $G'$, and $p$**:**
   **return** `true`;         // *Evidence for equivalence*
**else:**
   **return** `false`;         // *Definitely no equivalence*

$p$. Note that without this algorithm, even for $n = 4$, the problem of deciding inclusion of algebraic models required either manual computation with polynomials or extremely computationally expensive algorithms from algebraic geometry, so this algorithm is an enormous improvement.

### 3.3 TESTING MODEL EQUIVALENCE

Two graphs $G$ and $G'$ are called algebraically equivalent if $\overline{\mathcal{M}}(G) = \overline{\mathcal{M}}(G')$, which is the case iff $\overline{\mathcal{M}}(G) \subseteq \overline{\mathcal{M}}(G')$ and $\overline{\mathcal{M}}(G) \supseteq \overline{\mathcal{M}}(G')$. We can test both inclusions using Algorithm 2. But we can do a bit better by first checking if $G$ and $G'$ have the same skeleton, i.e. if each pair of nodes that is adjacent in $G$ is also adjacent in $G'$ and vice versa. By Corollary 7 in Section 4.1.1, for BAPs, having the same skeleton is a necessary condition for algebraic equivalence. Further, we realize that for BAPs, the dimension of the model is determined by the number of edges, and that if two different algebraic models have the same dimension, then neither can be contained in the other. So to decide equivalence of two BAPs with the same skeleton, it suffices to check inclusion in one direction.

We see immediately that Algorithm 3 has the same error probability and worst-case running time as Algorithm 2.

## 4 OTHER EQUIVALENCE RELATIONS ON GRAPHS

In this section, we discuss several different equivalence relations that have been considered in the literature to compare observational models $\mathcal{M}(G)$ of graphs $G$. We focus on how these equivalence relations compare to algebraic equivalence on BAPs, and what this means for the applicability of Algorithms 2 and 3 to the analogous decision problems for those equivalence notions.

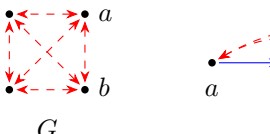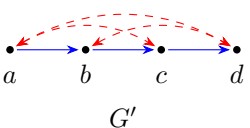

Figure 2: Two BAPs which are distributionally equivalent up to closure, but not distributionally equivalent, as $\mathcal{M}(G')$ excludes some covariance matrices that are present in $\mathcal{M}(G)$.

### 4.1 DISTRIBUTIONAL EQUIVALENCE

The most fine-grained equivalence relation that compares observational models is *distributional equivalence*. Graphs $G$ and $G'$ are called distributionally equivalent if $\mathcal{M}(G) = \mathcal{M}(G')$. This equivalence notion is considered for instance by Nowzohour et al. [2017].

Two graphs fail to be distributionally equivalent if even a single $\Sigma$ is present in $\mathcal{M}(G)$ but missing from $\mathcal{M}(G')$, or vice versa. Améndola et al. [2020] call $G$ and $G'$ *distributionally equivalent up to closure* if $\operatorname{cl}\mathcal{M}(G) = \operatorname{cl}\mathcal{M}(G')$, where $\operatorname{cl}\mathcal{M}(G)$ denotes the topological closure of $\mathcal{M}(G)$ in Euclidean topology. In other words, $\operatorname{cl}\mathcal{M}(G)$ contains $\mathcal{M}(G)$ and adds all points that are arbitrarily close to a point already in $\mathcal{M}(G)$.

The following theorem and example show how these two equivalence notions relate to algebraic equivalence for the case of BAPs.

**Theorem 5.** *For two BAPs $G$ and $G'$, $\operatorname{cl}\mathcal{M}(G) \subseteq \operatorname{cl}\mathcal{M}(G')$ iff $\overline{\mathcal{M}}(G) \subseteq \overline{\mathcal{M}}(G')$.*

*Proof.* Van Ommen and Mooij [2017] show that for HTC-identifiable $G$, almost all points in $\overline{\mathcal{M}}(G)$ are also in $\mathcal{M}(G)$. It follows that for BAPs, $\overline{\mathcal{M}}(G) = \operatorname{cl}\mathcal{M}(G)$, which proves the claim.                    □

An immediate consequence is that two BAPs are distributionally equivalent up to closure iff they are algebraically equivalent.

**Example 3.** The two graphs in Figure 2 are complete and hence impose no algebraic constraints. Since they are BAPs, it follows that they are distributionally equivalent up to closure. Yet they are not distributionally equivalent: the positive definite matrix

$$\Sigma = \begin{bmatrix} 1 & 3/4 & 2/9 & 1/2 \\ 3/4 & 1 & 3/4 & 1/2 \\ 2/9 & 3/4 & 1 & 1/2 \\ 1/2 & 1/2 & 1/2 & 1 \end{bmatrix},$$

is in $\mathcal{M}(G)$ but not in $\mathcal{M}(G')$. This can be seen by following the steps of the HTC-identification algorithm [Foygel et al., 2012]. This algorithm will successively compute $\lambda'_{ab}$,

$\lambda'_{bc}$ and $\lambda'_{cd}$ as solutions to systems of linear equations. For $\Sigma$, the first two systems have unique solutions, but the third has no solution. This proves that no parameter values $\Lambda', \Omega'$ exist for $G'$ such that $\phi(\Lambda', \Omega') = \Sigma$, so that $\mathcal{M}(G') \neq \mathcal{M}(G)$. Van Ommen and Mooij [2017, Figure 2] call a difference between $\mathcal{M}(G')$ and $\text{cl}\,\mathcal{M}(G')$ a *zero-measure constraint* and give an example for a graph that includes a bow; this example demonstrates such constraints can also occur among BAPs.

If two graphs $G$ and $G'$ are distributionally equivalent up to closure, then in practice it will not be possible to tell the difference based on finite data without further assumptions: if $\Sigma$ maximizes the likelihood in $\mathcal{M}(G)$, then $\Sigma'$'s will exist in $\mathcal{M}(G')$ that come arbitrarily close to this likelihood. Thus we argue that distributional equivalence (without 'up to closure') is too fine-grained for purposes of causal discovery, and distributional equivalence up to closure or coarser notions are more appropriate. If our definition of model $\mathcal{M}(\cdot)$ is believed to be reasonable in a particular setting (i.e., if the variables are real-valued, the relations linear, and higher-order moments can be ignored), then it follows from Theorem 5 that for causal discovery on BAPs, algebraic equivalence is the finest equivalence notion we could consider.

### 4.1.1 Graphical Conditions for Algebraic Equivalence

Nowzohour et al. [2017] show two necessary and one sufficient graphical conditions for distributional equivalence of two BAPs. The three criteria we show below are exactly analogous, but apply to algebraic rather than distributional equivalence. In these criteria, a *collider triple* is a triple $(u, v, w) \in V^3$ such that there is an edge between $u$ and $v$ as well as between $v$ and $w$, and both edges have an arrowhead at $v$. A *v-structure* is a collider triple where $u$ and $w$ are nonadjacent.

**Theorem 6** (Necessary condition). *Let $G$ and $G'$ be algebraically equivalent BAPs on vertex set $V$. Then for all $W \subseteq V$, the induced subgraphs $G_W$ and $G'_W$ are also algebraically equivalent.*

*Proof.* The proof of Nowzohour et al. [2017]'s Theorem 1 is built on theory from algebraic geometry, and can be seen to prove our claim without modification. A bit more specifically, the proof only considers the behaviour of the models near $\Sigma = I$, where $\mathcal{M}(G)$ and $\overline{\mathcal{M}}(G)$ coincide. We refer to Nowzohour et al. [2017] for the complete proof. $\square$

**Corollary 7.** *Two algebraically equivalent BAPs must have the same skeleton and v-structures.*

**Theorem 8** (Sufficient condition). *If two BAPs have the same skeleton and collider triples, they are algebraically equivalent.*

*Proof.* By Nowzohour et al. [2017]'s Theorem 2, two BAPs that satisfy this condition are distributionally equivalent, and distributional equivalence implies algebraic equivalence. $\square$

The conditions of Corollary 7 and Theorem 8 are easy to check by looking at the graphs and allow us to infer algebraic (non)equivalence of large sets of graphs without examining them one pair at a time. But they leave room between them: two BAPs that have the same skeleton and the same v-structures but different collider triples may or may not be algebraically equivalent. Establishing a single graphical criterion that is simultaneously necessary and sufficient for algebraic equivalence is an important open problem. Of course, for a specific pair of graphs, Algorithm 3 can be used to decide algebraic equivalence.

### 4.2 MARKOV AND NESTED MARKOV EQUIVALENCE

Two ADMGs are *Markov equivalent* if their models impose the same set of (conditional) independence constraints (or, in the context of LSEMs, vanishing (partial) correlation constraints). *Maximal ancestral graphs (MAGs)* [Richardson and Spirtes, 2002] are a special subclass of ADMGs for which the set of algebraic constraints and the set of (conditional) independence constraints are in one-to-one correspondence: by Corollary 8.19 of Richardson and Spirtes [2002], two MAGs $G$ and $G'$ impose the same set of (conditional) independence constraints iff $\mathcal{M}(G) = \mathcal{M}(G')$. Thus, when given two MAGs, Algorithm 3 decides whether they are Markov equivalent.

We slightly extend the result above to show that also Algorithm 2 can be used to compare Markov models when given two MAGs:

**Theorem 9.** *For two MAGs $G$ and $G'$, $\mathcal{M}_m(G) \subseteq \mathcal{M}_m(G')$ iff $\mathcal{M}(G) \subseteq \mathcal{M}(G')$ iff $\overline{\mathcal{M}}(G) \subseteq \overline{\mathcal{M}}(G')$.*

Here $\mathcal{M}_m(G)$ denotes the Markov model of $G$, i.e. the set of all distributions that satisfy all (conditional) independence constraints imposed by $G$.

*Proof.* $\mathcal{N}$ denotes the set of all Gaussian distributions, and here we will regard $\mathcal{M}(G)$ as the set of all Gaussian distributions in the LSEM model of $G$ (instead of as the set of all covariance matrices of those distributions as we do elsewhere).

First, we claim that $\mathcal{M}_m(G) \cap \mathcal{N} \subseteq \mathcal{M}_m(G') \cap \mathcal{N}$ iff $\mathcal{M}_m(G) \subseteq \mathcal{M}_m(G')$. The proof is analogous to that of Theorem 8.13 of [Richardson and Spirtes, 2002]: First, the implication from right to left is obvious. For the other direction, suppose $\mathcal{M}_m(G) \cap \mathcal{N} \subseteq \mathcal{M}_m(G') \cap \mathcal{N}$. By Theorem 7.5 of Richardson and Spirtes, there exists a distribution $N \in \mathcal{N}$

faithful to $\mathcal{M}_m(G)$. This $N$ is also in $\mathcal{M}_m(G') \cap \mathcal{N}$. It follows that any (conditional) independence imposed by $G'$ is also imposed by $G$; i.e., $\mathcal{M}_m(G) \subseteq \mathcal{M}_m(G')$.

By Theorem 8.14 of Richardson and Spirtes, for a MAG $G$, $\mathcal{M}(G) = \mathcal{M}_m(G) \cap \mathcal{N}$. So $\mathcal{M}(G) \subseteq \mathcal{M}(G')$ iff $\mathcal{M}_m(G) \cap \mathcal{N} \subseteq \mathcal{M}_m(G') \cap \mathcal{N}$, which by the claim above is equivalent to $\mathcal{M}_m(G) \subseteq \mathcal{M}_m(G')$. Since also $\overline{\mathcal{M}}(G) = \mathcal{M}_m(G) \cap \mathcal{N}$, the claim about $\overline{\mathcal{M}}$ follows. $\square$

For any graph, we can define its algebraic model and see which algebraic constraints it imposes. Some of these constraints may correspond to (conditional) independences, but others may be of the more general kinds listed on page 3, which are ignored by Markov equivalence. Thus for general graphs, Markov equivalence is coarser than algebraic equivalence, so that using algebraic equivalence in causal discovery will give us more power to distinguish between different graphs than the more commonly used Markov equivalence gives us. This is what motivated us to research algebraic equivalence in this paper.

*Nested Markov equivalence* [Shpitser et al., 2014, Richardson et al., 2023] refines ordinary Markov equivalence by considering not only (conditional) independences in the observational distribution, but also in kernels. These kernels can be understood as representing interventional distributions that can be identified from the observational distribution. For example, for the graph in Figure 1(a), the distribution after intervening on $X_b$ is identifiable, and in this distribution, given $X_c$, the value of $X_d$ is independent of that of $X_b$. This conditional independence in a kernel translates back to a constraint on the original observational distribution: a *nested Markov constraint*.

Like ordinary Markov equivalence, but unlike algebraic equivalence and distributional equivalence (up to closure), nested Markov equivalence does not depend on the ranges of the random variables or on parametric assumptions such as linearity. It does have a special role in the context of discrete variables: as shown by Evans [2018], the nested Markov model reflects all equality constraints on the observed distribution. Thus it is to discrete variable models as the notion of algebraic equivalence studied in this paper is to LSEMs.

Shpitser et al. [2018] define a subclass of BAPs called *maximal arid graphs (MArGs)*, as well as a projection operator that takes any ADMG $G$ to a nested Markov equivalent MArG $G^\dagger$. In a MArG, each nonadjacency corresponds to a nested Markov constraint. As such, MArGs play the same role for nested Markov models as MAGs play for ordinary Markov models.

In the following theorem, $\mathcal{M}_n(G)$ denotes the nested Markov model of $G$: the set of all distributions that satisfy all nested Markov constraints imposed by $G$.

**Theorem 10.** *For two MArGs $G$ and $G'$, if $\mathcal{M}_n(G) \subseteq \mathcal{M}_n(G')$ then $\mathcal{M}(G) \subseteq \mathcal{M}(G')$ (and thus $\overline{\mathcal{M}}(G) \subseteq \overline{\mathcal{M}}(G')$).*

*Proof.* As in the proof of Theorem 9, let $\mathcal{M}(G)$ denote the set of all Gaussian distributions in the LSEM model of $G$. By Shpitser et al. [2018, Theorem 35], $\mathcal{M}(G) = \mathcal{M}_n(G) \cap \mathcal{N}$ for any MArG $G$. So $\mathcal{M}_n(G) \subseteq \mathcal{M}_n(G') \Rightarrow \mathcal{M}(G) = \mathcal{M}_n(G) \cap \mathcal{N} \subseteq \mathcal{M}_n(G') \cap \mathcal{N} = \mathcal{M}(G')$. $\square$

In other words, inclusion of one algebraic model in another is a necessary condition for the corresponding inclusion of nested Markov models. This means that Algorithms 2 and 3 can be used to establish that certain pairs of graphs are not nested Markov equivalent.

We conjecture that also the converse implication holds. We verified this empirically on all MArGs of up to five nodes, by fitting algebraically equivalent MArGs on random discrete data and checking that the attained likelihood scores were close. We used the maximum likelihood fitting procedure described by Evans and Richardson [2010, 2019], as implemented in `Ananke` [Lee et al., 2023]. If this conjecture is true, it would follow that Algorithms 2 and 3 can also be used to decide inclusion and equivalence of nested Markov models, by first applying the maximal arid projection to the input graphs.

## 5 EXPERIMENTAL RESULTS

To demonstrate the practical usability of our algorithms, we conducted a small experiment, measuring the running time and number of errors of Algorithm 2 on graphs of $n$ vertices and different primes $p$. As inputs, we used all pairs from the family of graphs that appear in the proof of Lemma 4 (Appendix B.3) as the maximizers of the error probability bound among all graphs of that size. These graphs should also maximize the running time, as they require `solve(v)` to be called on all vertices $v$. These results are for our Python implementation; see Appendix C for details.

The results are displayed in Table 1. Clearly, larger graphs

Table 1: Average running time, number of false positives (out of at least 2000 non-inclusion instances), and theoretical upper bound on the probability of error of Algorithm 2, for graphs of $n$ vertices that maximize this bound, using prime $p$ as a modulus.

| $n$ | $p$ | time (ms) | #FP | error bound |
|---|---|---|---|---|
| 5 | $2^{31} - 1$ | 9.23 | 0 | $4.61 \cdot 10^{-8}$ |
| 25 | $2^{31} - 1$ | 682 | 0 | 0.287 |
| 25 | $2^{63} - 25$ | 663 | 0 | $6.68 \cdot 10^{-11}$ |
| 25 | $2^{127} - 1$ | 680 | 0 | $3.62 \cdot 10^{-30}$ |

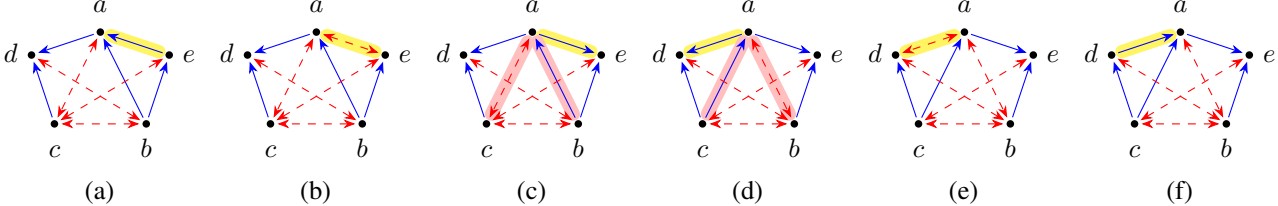

Figure 3: An algebraic equivalence class consisting of six BAPs. Graphs (a–c) differ by one edge (highlighted in yellow) and the same is true for (d–f). But between these two clusters, the difference is at least two edges (highlighted in pink).

increase the computation time, while $p$ seems to have little impact. For the two bottom rows, we resort to Python's big-integer arithmetic, but this does not lead to a performance penalty here. This suggests that if it is necessary to reduce the probability of error, it is better to increase $p$ rather than run the algorithm repeatedly (though this may be implementation-dependent). Also noteworthy is that the algorithm never returned a wrong result. For most rows, this was to be expected as the probability of error is known to be extremely small in those cases. But for $n = 25$ and $p = 2^{31} - 1$, the bound would have allowed many hundreds of false positives while none were observed, demonstrating that the actual probability of error in this case is much smaller than the bound suggests.

The only other algorithm for deciding algebraic equivalence with runtime comparable to ours is the empirical equivalence test used by Nowzohour et al. [2017]. To compare these algorithms, we did an experiment using their code and recommended settings to score all 543 complete BAPs on 4 nodes (i.e. BAPs with an edge of some type between each pair of nodes), using data randomly sampled from the BAP model with six bidirected edges. These graphs are all algebraically equivalent, yet the empirical equivalence test incorrectly concludes that over 80% of pairs are not equivalent on average. In contrast, our algorithm provably returns 'true' for a pair of algebraically equivalent graphs.

## 6 DISCUSSION AND FUTURE WORK

The algorithms presented in this paper are the first that can efficiently reveal the relation between the algebraic models for any pair of BAPs. Unfortunately, they do not immediately provide insight into the contents of an algebraic equivalence class. One might hope that by starting from some graph $G$ and repeatedly making local changes to it, checking (with Algorithm 3) each time that the resulting graph is algebraically equivalent to $G$, one will find a list containing all graphs in $G$'s algebraic equivalence class. A natural choice for such a local change operation would be to replace any edge between $v$ and $w$ with another type of edge [Nowzohour et al., 2017]. But as we see in Figure 3, we may recover only part of an equivalence class this way.

Markov equivalence can be graphically characterized for DAGs in terms of the skeleton and v-structures [Verma and Pearl, 1991], and for the more general ancestral graphs in terms of the skeleton and 'colliders with order' [Ali et al., 2009, Claassen and Bucur, 2022]. For algebraic equivalence, separate necessary and sufficient graphical conditions exist (see Section 4.1.1 for BAPs, or [Van Ommen and Mooij, 2017, Theorem 2] for more general graphs), but no characterization that is simultaneously necessary and sufficient (except in MAGs, where it coincides with Markov equivalence). Such a characterization would be a step towards an analogue of CPDAGs and PAGs, which are graphs that represent entire equivalence classes. This would solve problems such as the one seen in Figure 3, and would be the most suitable format for a causal discovery algorithm's output.

Other future work is to extend our algorithms beyond BAPs to more general graphs.

## 7 CONCLUSION

We have introduced the first efficient algorithms for the tasks of determining whether a graph imposes a given algebraic constraint, whether the algebraic model of one graph is a submodel of another, and whether two graphs have the same algebraic model. We argue that for linear, possibly Gaussian models, algebraic equivalence is the most appropriate equivalence notion that causal discovery algorithms can use. We conjecture that algebraic equivalence can be related to nested Markov equivalence, which would also make our algorithms applicable to the discrete and nonparametric cases.

### Acknowledgements

I want to thank all reviewers, whose careful reading and valuable suggestions substantially improved the presentation of this work.

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

# Efficiently Deciding Algebraic Equivalence of Bow-Free Acyclic Path Diagrams (Supplementary Material)

**Thijs van Ommen**[1]

[1]Information and Computing Sciences, Utrecht University, Utrecht, The Netherlands

## A    MORE DETAILS ABOUT EXAMPLES 1 AND 2

In this appendix, we provide evidence for the claims made in Examples 1 and 2, and include some further discussion.

### A.1    EXAMPLE 1

The constraint construction algorithm of Van Ommen and Drton [2022] requires as input a sequence of sets $(Y_v)_v$ satisfying certain properties outlined by Foygel et al. [2012]. We use $Y_v = \mathrm{pa}(v)$ for all $v \in V$. This choice is valid for all BAPs and is used throughout this paper when applicable.

The matrix $\Sigma$ in the example was found by first using a computer algebra package to compute the primary decomposition of the graphically represented ideal. This reveals that the ideal has multiple components: the component describing the model and fifteen spurious components. Most of the spurious components have a principal minor of $\Sigma$ as one of their generators, and thus describe sets of $\Sigma$'s on the boundary of the positive definite cone. One spurious component does allow $\Sigma$'s inside the positive definite cone:

$$\left\langle \sigma_{ae}, \sigma_{be}, \sigma_{ce}, \begin{vmatrix} \sigma_{bd} & \sigma_{bc} \\ \sigma_{cd} & \sigma_{cc} \end{vmatrix}, \begin{vmatrix} \sigma_{aa} & \sigma_{ab} & 0 \\ \sigma_{ba} & \sigma_{bb} & \sigma_{bd} \\ \sigma_{ca} & \sigma_{cb} & \sigma_{cd} \end{vmatrix}, \begin{vmatrix} \sigma_{aa} & \sigma_{ab} & 0 \\ \sigma_{ba} & \sigma_{bb} & \sigma_{bc} \\ \sigma_{ca} & \sigma_{cb} & \sigma_{cc} \end{vmatrix} \right\rangle.$$

For the $\Sigma$ given in Example 1, all generators above are 0 — for the first five generators, this can be seen by simply filling in the zero entries of $\Sigma$; for the final generator, the determinant equals $1 - \frac{9}{16} - \frac{9}{16} + \frac{2}{16} = 0$.

The HTC-identification algorithm requires taking the inverse of the $3 \times 3$ matrix that appears in the final generator. Spurious components of the graphically represented ideal may arise in places where such an inverse fails to exist, as is the case here. The matrix resembles a principal minor of $\Sigma$, except that one of its entries has been replaced by a zero. If it had been a principal minor, then the HTC-identification algorithm would have been able to take its inverse for all positive definite $\Sigma$. While this ideal is not PD-primary, it does have the weaker property of being $I$-primary, because the matrix in question is invertible at $\Sigma = I$.

### A.2    EXAMPLE 2

This graph is not a BAP, so the choice $Y_v = \mathrm{pa}(v)$ is not valid. To establish that the graph is HTC-identifiable, we can choose $Y_a = \varnothing, Y_b = \{a\}, Y_c = \{a, b\}, Y_d = \{a\}, Y_e = \{a, d\}$. The results below are for the graphical ideal obtained using this choice of the $(Y_v)_v$ as input to the constraint construction algorithm; other choices are possible and lead to similar results.

As for Example 1, we computed the primary decomposition of the graphical ideal using a computer algebra package. We find that one of the spurious components is simply $\langle \sigma_{ac}, \sigma_{ad} \rangle$. This component admits the identity matrix, establishing that this ideal is not $I$-primary.

We will write $G'$ to refer to the graph in Figure 1(b). While the graphically represented ideal fails to describe $\overline{\mathcal{M}}(G')$ accurately, an accurate description of the algebraic model can be obtained using the theory of Fink et al. [2016]. If a bidirected edge $d \leftrightarrow e$ is added to $G'$, we obtain a new graph $G'^+$ that is algebraically equivalent to $G'$ [Van Ommen and Mooij, 2017, Theorem 2]. In the terminology of Fink et al. [2016], $G'^+$ is a *generalized Markov chain*, for which they show that the vanishing minor constraints implied by t-separation correctly generate the ideal of $\overline{\mathcal{M}}(G'^+)$, and thus of $\overline{\mathcal{M}}(G')$. These generators are $\langle |\Sigma_{ab,cd}|, |\Sigma_{ab,ce}|, |\Sigma_{ab,de}| \rangle$. However, any graphically represented ideal of $G'$ has only two generators, which is a way to understand why the graphically represented ideal has problematic spurious components. This also shows that we can test whether some $\overline{\mathcal{M}}(G)$ is contained in $\overline{\mathcal{M}}(G')$ by running Algorithm 1 on $G$ three times: once for each of the three generators listed above.

# B ADDITIONAL PROOFS

## B.1 PROOF OF THEOREM 1

*Proof.* For each spurious component $K$ in the primary decomposition of $J$, $\overline{\mathcal{M}}(G) \cap V(K) \subsetneq \overline{\mathcal{M}}(G)$, as the identity matrix $I \in \overline{\mathcal{M}}(G)$ but $I \notin V(K)$. Because $\overline{\mathcal{M}}(G)$ is an irreducible variety [Cox et al., 2015] and the intersection $\overline{\mathcal{M}}(G) \cap V(K)$ is an algebraic variety, the latter, and hence $\mathcal{M}(G) \cap V(K)$, must be of lower dimension than $\overline{\mathcal{M}}(G)$. $\mathcal{M}(G) \cap V(J)$ is the union of a finite number of such intersections and of the non-spurious part $\mathcal{M}(G) \cap \overline{\mathcal{M}}(G')$. It follows that $\mathcal{M}(G) \cap V(J) \setminus \overline{\mathcal{M}}(G')$ is also of lower dimension than $\mathcal{M}(G)$, which is of the same dimension as $\overline{\mathcal{M}}(G)$. $\qquad\square$

## B.2 PROOF OF THEOREM 3

*Proof.* First note that for each pair $\{v, w\}$ of nonadjacent nodes in $G'$, the value of $\tilde{\Omega}'_{vw}$ computed by the algorithm equals the evaluation of the graphically represented constraint of Van Ommen and Drton [2022] at $\Sigma$. For the $(Y_v)_v$ that are needed as input to the constraint construction algorithm, we use $Y_v = \mathrm{pa}_{G'}(v)$ for all $v \in V$: this choice is valid for all BAPs. Both computations follow the half-trek identification algorithm of Foygel et al. [2012], with one exception: when $\Lambda_{\cdot,v}$ is computed, Cramer's rule is used to show that $|\mathbf{A}^{(v)}| \cdot [I - \Lambda]_{\cdot,v} = [|\mathbf{A}^{(v)}|, |\mathbf{A}^{(v)}_{w_1}|, \dots, |\mathbf{A}^{(v)}_{w_k}|]$ for $\mathrm{pa}_{G'}(v) = \{w_1, \dots, w_k\}$, but the $|\mathbf{A}^{(v)}|$ is not divided out.

If $\overline{\mathcal{M}}(G) \subseteq \overline{\mathcal{M}}(G')$, then any $\Sigma \in \mathcal{M}(G) \subseteq \overline{\mathcal{M}}(G)$ will satisfy any algebraic constraint that holds in $\overline{\mathcal{M}}(G')$. In particular, it will satisfy $\tilde{\Omega}'_{v,w} = 0$ for all $\{v, w\}$ nonadjacent in $G'$. The algorithm will always return 'true' in this case.

For the case $\overline{\mathcal{M}}(G) \nsubseteq \overline{\mathcal{M}}(G')$, we will have to account for the possibility that the graphically represented ideal $J$ may have spurious components, so that $V(J) \supseteq \overline{\mathcal{M}}(G')$. As shown by Van Ommen and Drton [2022], for acyclic graphs, $\Sigma \in V(J) \setminus \overline{\mathcal{M}}(G')$ implies that for some $v \in V$, the polynomial $|\mathbf{A}^{(v)}|$ evaluates to zero at $\Sigma$. Van Ommen and Drton further show that if $G'$ is bow-free, $|\mathbf{A}^{(v)}|$ evaluates to 1 at $\Sigma = I = \phi(\mathbf{0}, I)$ (i.e. the graphically represented ideal is $I$-primary). Thus it is not the zero polynomial in terms of $(\Lambda, \Omega)$.

Having ruled out the possibility that $\tilde{\Omega}_{vw} \circ \phi \equiv 0$ for all $\{v, w\}$ nonadjacent due to $\overline{\mathcal{M}}(G)$ being contained in a spurious component of $V(J)$, we conclude that an $\tilde{\Omega}_{vw} \circ \phi$'s being identically zero must imply that $\overline{\mathcal{M}}(G) \subseteq \overline{\mathcal{M}}(G')$. Equivalently, $\overline{\mathcal{M}}(G) \nsubseteq \overline{\mathcal{M}}(G')$ implies that for some nonadjacent $\{v, w\}$, $\tilde{\Omega}_{v,w}$ is not the zero polynomial.

Considered as polynomials over $\Sigma$, we see by induction that the entries of $\mathbf{M}^{(v)}_{w,\cdot}$ in `solve`$(v)$ have degree at most $a_w$ if $w \in \mathrm{htr}_{G'}(v)$ and 0 otherwise; the entries of $\mathbf{A}^{(v)}_{w,\cdot}$ and $\mathbf{b}^{(v)}_w$ have degree at most $a_w + 1$ if $w \in \mathrm{htr}_{G'}(v)$ and 1 otherwise; and the determinant $|\mathbf{A}^{(v)}|$ and the entries of $\Lambda_{\cdot,v}$ have degree at most $a_v$. Then $\deg \tilde{\Omega}'_{v,w} \leq a_v + a_w + 1$.

Now, similar to the dimension argument of Theorem 1 but using the Schwartz–Zippel lemma as in the proof of Theorem 2, the probability of error is bounded by

$$P[\tilde{\Omega}(\phi(\Lambda, \Omega))_{vw} = 0 \mid \tilde{\Omega}_{vw} \circ \phi \not\equiv 0] \leq \frac{1}{p}(2\ell_G + 1)(a_v + a_w + 1).$$

Because we do not know for which $\{v, w\}$ the constraint is not the zero polynomial, we take the maximum over all candidates.

Interestingly, if the algorithm encounters an $|\mathbf{A}^{(v)}|$ that evaluates to zero but also a nonzero $\tilde{\Omega}_{v,w}$ for $\{v, w\}$ nonadjacent, then it can and will report 'false'. Thus this case does not contribute to the error probability.

All operations outside `solve()` can clearly be performed in $O(n^\omega)$ time. Within `solve()`, $\text{htr}_{G'}(v)$ can be computed by breadth-first search in $O(n^2)$, and $\mathbf{A}^{(v)}$ and $\mathbf{b}^{(v)}$ can be computed in $O(n^\omega)$. Write $k = |\text{pa}_{G'}(v)|$. Computing $\tilde{\Lambda}'_{\cdot,v}$ in the final line involves the computation of $k+1$ determinants, namely the $k \times k$ minors of a $k \times (k+1)$ matrix. Like matrix multiplication, determinants can be computed in time $O(n^\omega)$ [Bunch and Hopcroft, 1974], and we can use the technique of Baur and Strassen [1983] to compute all $k+1$ minors still in time $O(n^\omega)$ (though in our implementation, we used an approach based on Gaussian elimination that runs in time $O(n^3)$; see Appendix C). In the worse case, `solve()` is performed $n$ times, making the time complexity of Algorithm 2 $O(n^{\omega+1})$. $\qquad\square$

## B.3 PROOF OF LEMMA 4

*Proof.* The hard part is bounding the degree of the algebraic constraint, over all possible BAPs $G'$. Assume the nodes of $G'$ are topologically ordered. We want to find numbers $a_v$ such that for any BAP $G'$, $\deg|\mathbf{A}^{(v)}| \le a_v$ for all $v$ for which the algorithm calls `solve`.

First, $a_1 = 0$ since node 1 has no parents, and $a_2 = 1$ since node 2 may only have node 1 as a parent.

For $v \ge 3$, we could have $\text{pa}(v) = \{1, 2, \ldots, v-1\}$, in which case no half-treks exist from $v$ to any of these parents, and we would have $\deg|\mathbf{A}^{(v)}| = v-1$. By including a single bidirected edge between nodes 1 and $v$ taking $\text{pa}(v) = \{2, 3, \ldots, v-1\}$, all such half-treks might exist, so $a_v = v - 2 + \sum_{i=2}^{v-1} a_i$. A direct expression is $a_v = 3 \cdot 2^{v-3} - 1$ (for $v \ge 3$).

There must be a pair of nonadjacent nodes in $G'$ for it to impose an algebraic constraint. Let $s$ and $t$ be two nonadjacent nodes, with $s < t$. Then the bound $a_s$ is computed as above, but $\deg|\mathbf{A}^{(t)}|$ will obey a tighter bound, because it must have fewer adjacencies to earlier nodes than used in the argument above. We want to establish an upper bound $a'_t$ to this degree. Assume $t \ge 4$. If $s \ne 1$, we still want a bidirected edge between nodes 1 and $t$, in which case we would get $a'_t = t - 3 + \sum_{i=2}^{t-1} a_i - a_s$. If $s = 1$, the bidirected edge would go between nodes 2 and $t$, and $a'_t = t - 3 + \sum_{i=2}^{t-1} a_i - a_2$.

The bound on the degree of the algebraic constraint is $1 + a_s + a'_t$. Since the sequence $a_1, a_2, \ldots$ is increasing, this is maximized when $t = n$. If $n \ge 4$, all choices for $s \ge 2$ yield the same value, because $a_s$ is both subtracted and added; $s = 1$ yields one less because $a_2 = 1$ is subtracted and $a_1 = 0$ is added.

For these choices of $s$ and $t$ and for $n \ge 4$, the degree bound becomes

$$1 + a_s + a'_n = 1 + a_s + n - 3 + \sum_{i=2}^{n-1} a_i - a_s = n - 2 + \left(1 + \sum_{i=3}^{n-1} a_i\right) = \sum_{i=3}^{n-1}(3 \cdot 2^{i-3} - 1) + n - 1$$

$$= 3\sum_{i=0}^{n-4} 2^i - (n-3) + n - 1 = 3(2^{n-3} - 1) + 2 = \frac{3}{8}2^n - 1. \quad\square$$

## C IMPLEMENTATION

The algorithms described in this paper are implemented in Python using the Galois library [Hostetter, 2020] for computations over $\mathbb{F}_p$. The experiments in Section 5 were performed with Python 3.11, NumPy version 1.26.4, and Galois version 0.3.8, on a MacBook Pro (2.3 GHz Intel processor).

Algorithm 2 requires the computation of all $n \times n$ minors of an $n \times (n+1)$ matrix, namely the matrix $\mathbf{A}^{(v)}$ augmented with the column vector $\mathbf{b}^{(v)}$. We use the following implementation to perform this computation in $O(n^3)$ time, i.e. the same complexity as computing a single determinant using Gaussian elimination. First, applying Gaussian elimination to the augmented matrix allows us to find the determinant of $\mathbf{A}^{(v)}$, as well as the minor obtained by omitting the second-to-last column. Then we imagine we flip the matrix left-to-right, so that the nonzero elements now reside in the top left triangle. Next we apply Gaussian elimination to the bottom two rows of this flipped matrix; ignoring the third column and using that the other columns form a permuted triangular matrix, we compute the third minor. Each subsequent minor is computed in this fashion, with the final minor requiring a Gaussian elimination of the entire flipped matrix. The successive Gaussian eliminations on the flipped matrix benefit from the fact that the previous iteration already put the matrix in close-to-triangular form, and that most of the rows they operate on are known to be largely zeros, so that they together require only roughly half as many operations as the initial Gaussian elimination.