# OpenReview forum: "Efficiently Deciding Algebraic Equivalence of Bow-Free Acyclic Path Diagrams"
_auai.org/UAI/2024/Conference — UAI 2024 poster_

### Official Review · Reviewer_jLX2 · 2024-03-07

**Q2-1 Originality-Novelty:** 2
**Q2-2 Correctness-Technical Quality:** 4
**Q2-5 Clarity Of Writing:** 3

**Q1 Summary And Contributions:**

The paper proposes an algebraic approach for deciding the algebraic equivalence of BAP diagrams. The algorithm is based on finite field computations, which may lead to incorrect decisions; an upper bound on the probability of this error is provided.

**Q2-3 Extent To Which Claims Are Supported By Evidence:**

2: Fair: the main claims are somewhat supported by evidence (but the experimental evaluation may be weak, or does not match entirely with the claims, important baselines may be missing, proofs contain important ideas but lack rigor, algorithmic details are only discussed superficially, references are imprecise, assumptions are not sufficiently motivated or explicated, etc.).

**Q2-4 Reproducibility:**

2: Fair: key resources (e.g. proofs, code, data) are unavailable but key details (e.g. proof sketches, experimental setup) are sufficiently well-described for an expert to confidently reproduce the main results.

**Q3 Main Strengths:**

- The problem studied is relevant to the community, and a complete solution could lead to important future developments. The paper is well organized and easy to read even although the use of vocabulary from Algebraic Geometry.
- The explicit use of “mod p” computation is interesting and I have never seen it in this type of literature.

**Q4 Main Weakness:**

- In my opinion, the main result is too weak. It is not clear how strong the evidence for the equivalence would be in case the algorithm returns “TRUE”. A computation study for small graphs comparing the outcome of the proposed algorithm with the one from a Grobner Basis computation (or something similar) is necessary.
- The proof of Theorem 6 must be explained, as it is now is confusing to a reader that is not familiar with AG.

**Q5 Detailed Comments To The Authors:**

As I mentioned in Q4, I think it would be necessary to implement the algorithm to actually show to what extent it is feasible to use it, and how informative it can be.

Here you can find a list of small adjustments/typos

- Abstract: "two pairs of graphs"—> two graphs/ a pair or graphs.
- The definition of an Ideal is missing, also the definition of a trek is vague. It would be useful to have a section in the appendix with these formal definitions.
- The authors should stress that in the non-Gaussian case, the two BAPs are never equivalent (using higher order moments), and refer to [Wang and Drton, 2023, JMLR] for the proof.

**Q9 Complying With Reviewing Instructions:**

Yes

---

> ### Author Rebuttal · Authors · 2024-04-03
>
> Thank you for your positive comments about the paper's relevance, clarity and novelty, and for your other remarks, which we will address when revising the paper. Thanks in particular for pointing out the reference [Wang and Drton, 2023, JMLR], which is indeed pertinent in the context of this paper.
>
> You mention two concerns about the paper, which we hope to address below:
>
> * "In my opinion, the main result is too weak. It is not clear how strong the evidence for the equivalence would be in case the algorithm returns “TRUE”. A computation study for small graphs comparing the outcome of the proposed algorithm with the one from a Grobner Basis computation (or something similar) is necessary."
>
> The strength of evidence is quantified by Theorems 2 and 3 (and Lemma 4), which give upper bounds to the probability of returning 'true' when the correct answer is 'false' (which is the only possible kind of error). As indicated below Lemma 4, this probability is at most $4.61 \cdot 10^{-8}$ for small graphs ($n=5$), meaning that a 'true' from the algorithm is extremely strong evidence for equivalence. Also for large graphs, a small error probability can always be ensured, at the cost of more computation time.
>
> To illustrate this, we will add a computational study to the paper. This will reinforce the message that the error probability is really small, and it will also provide concrete insight into the computation times.
>
> * "The proof of Theorem 6 must be explained, as it is now is confusing to a reader that is not familiar with AG."
>
> We could add a more detailed argument in the appendix to explain why the steps of Nowzohour et al. [2017]'s proof carry over to algebraic equivalence. However, this probably would not make the proof clear for a reader not familiar with algebraic geometry. Do you maybe have a different suggestion how we could address this concern? E.g. is there some aspect of the theorem itself that could be clarified?

---

### Official Review · Reviewer_114P · 2024-03-14

**Q2-1 Originality-Novelty:** 2
**Q2-2 Correctness-Technical Quality:** 3
**Q2-5 Clarity Of Writing:** 1

**Q10 Ethical Concerns:**

No.

**Q1 Summary And Contributions:**

This paper introduces algorithms to judge whether two bow-free acyclic graph imply the same algebraic constraints in the context of (Gaussian) linear structural equation models. The authors argue that algebraically equivalent graphs are also not distinguishable from data hence it would be not worth computing individual scores for both. The algorithm randomly samples coefficient and covariance matrices that could come from such graphs. If equivalence holds indeed, these must fulfil the constraints such that the algorithm always claims equivalence. Otherwise the constraints are not fulfilled with high probability such that non-equivalence is outputted. The error can be controlled using suitable computational effort.

**Q2-3 Extent To Which Claims Are Supported By Evidence:**

2: Fair: the main claims are somewhat supported by evidence (but the experimental evaluation may be weak, or does not match entirely with the claims, important baselines may be missing, proofs contain important ideas but lack rigor, algorithmic details are only discussed superficially, references are imprecise, assumptions are not sufficiently motivated or explicated, etc.).

**Q2-4 Reproducibility:**

2: Fair: key resources (e.g. proofs, code, data) are unavailable but key details (e.g. proof sketches, experimental setup) are sufficiently well-described for an expert to confidently reproduce the main results.

**Q3 Main Strengths:**

The paper seems to be mathematically solid. I did not see any flaws (I did not read the proofs in the appendix).
The algorithms are constructed in a clever way, i.e., random sampling to check the constraint. I especially like that a user can flexibly control the trade-off between computational investment and error probability.

**Q4 Main Weakness:**

1. The focus on bow-free acyclic graphs is not justified. It seems to be mainly for mathematical convenience as many of the results assume this. But, there is no argument why this bow-freeness could be reasonable if we receive an ADMG in practice.

2. The paper is hard to read and not self-contained. In the introduction, many terms are included without properly introducing and defining them. Although background references are given in the theory part, it would take me a lot of additional effort invested in reading these to value the authors' contributions. Also, references to specific definitions should be more precise.

3. As the main selling point in the title is ``efficiently deciding'' this should at least be practically demonstrated. I.e., for some data example - simulated if you wish - one should see how this is better / more efficient than solely relying on score-based algorithms.

4. The authors argue that these algebraic constraints are the most fine-grained resolution and hence more powerful than the independence constraints for linear structural equations models. But, they should explain more clearly whether they have any relevance for broader model classes. This could potentially increase the impact of the new algorithms.

**Q5 Detailed Comments To The Authors:**

- p1: ``superexponential size of the search space''. Which search space? This is not defined.

- p1: ``For a graph without latent variables, its statistical model can be fully described by a list of (conditional) independences
that must hold between the variables.'' This is not generally true. You should at least mention the Markov assumption.

- p1: You use 'BAP' before properly introducing it as an abbreviation in Section 2.

- p1: ``An implementation of these algorithms will be made public if the paper is published.'' If you consider your work relevant, you should provide these algorithms regardless of the outcome here.

- p2: ``Scoring a graph is an expensive operation requiring iterative optimization algorithms, and the result is not reliable due to numerical inaccuracy.'' Does this hold even under the linearity assumption?

- p2: ``None of these methods can be used to decide whether one model is included in another''. What does it mean for a model to include another?

- p2: ``for $ v\neq w$ must have $Cov(\epsilon_v, \epsilon_w = 0)$ unless there is a bidirected edge between $v$ and $w$'' Do you then also assume independence? This would typically be the case in such causal models.

- p2: ``If G contains a directed cycle, we also need to require that $(I-\Lambda)$ is invertible.'' Do you even consider directed cycles at some point? Does this mean that invertibility is implied without directed cycles?

- p2: Are the directed edges in half-treks also meant to be traversed in the backward direction? From the examples later, it seems so. This should be more precise.

- p2: There is no intuition for the relevance of treks and half-treks.

- p2: ``A graph is HTC-identifiable sets $Y_v$ exist for each $v \in V$ satisfying certain properties.'' This is not helpful for the understanding. Also, is there a missing word in this sentence?

- p3: ``A useful simplification is to keep only the equalities that are true for all $\Sigma$''. Does this not hold for all constraints? If a constraint is not fulfilled by all $\Sigma$, how could these be part of $M(G)$?

- p3: ``The polynomial is simply $\sigma_{vw}$. For multivariate Gaussians, $\sigma_{vw} = 0$ is equivalent to marginal independence.'' For other linear structural equation models, this is also true up to term cancellations?

- p3: 'ideal' is not defined.

- p3: 'positive definite cone' is introduced without context.

- p3: 'primary decomposition' is not defined. Hence, also the example is not helpful. We just get to see a matrix.

- p3: The paragraph from ``If a bidirected edge ... '' brings several new terms without explanation. This is not helpful.

- p4: $V(J)$ is not defined

- p4: $\mathbb{F}_p$ is not defined.

- p4: What is the standalone value of Algorithm 1 if it is not even used as an input to Algorithm 2? All the discussion is about comparing graphs. Why would one want to compare a graph to an individual constraint?

- p5: Footnote 3 is rather irrelevant. If space is needed to explain other concepts better, I would remove it.

- p5: I am not sure if I understand model inclusion here correctly. $\overline{M}(G) \subseteq \overline{M}(G')$ means that $G$ satisfies at least all constraints from $G'$. As $G'$ is a BAP, we can calculate these and test for them. Correct? I suggest making this more explicit in the text.

- p5: ``$w \in pa_{G'} (v)$ with $w \in htr_{G'} (v)$''. Couldn't you write this as intersection $w \in pa_{G'} (v) \cap htr_{G'} (v)$?

- p6: Can you give intuition on the concept of closure? Graphs are discrete objects, how can they be arbitrarily close?

- p7: ``Markov equivalence is coarser than algebraic equivalence, as it considers only (conditional) independence constraints, which form a strict subset of algebraic constraints.'' Is this justified somewhere?

- p7: The section on MArGs comes without a proper introduction to the concepts, hence it is very hard to follow.

- p8: ``This means that Algorithms 2 and 3 can be used to establish that certain pairs of graphs are not nested Markov equivalent.'' Does this mean that you can test equivalence relations in general models satisfying the Markov property and not just in LSEMs? If yes, I think this deserves a much more prominent mention as it widens the scope a lot.

**Q9 Complying With Reviewing Instructions:**

Yes

---

> ### Author Rebuttal · Authors · 2024-04-04
>
> We would like to thank the reviewer for their thorough review and many useful comments. We acknowledge that parts of the submitted manuscript can be hard to follow, as they build on a diverse range of earlier literature. With the additional space available for the camera-ready version and the knowledge of which parts of the text caused confusion, we believe we can revise the paper so that the core ideas of the paper will be clearly communicated in a self-contained manner.
>
> Below we respond to the four items you listed as weaknesses of the paper.
>
> 1. [reasons for the limitation to bow-free acyclic graphs]
>
> The restriction to BAPs is indeed primarily for mathematical reasons. It is a restriction that is somewhat common in the literature, as graphs with bows and/or directed cycles have properties that make them difficult to analyse from an algebraic point of view. For example, it is often impossible to identify the parameters of such models. For our present work, Example 2 showcases the relevant problem: we do not have an efficient automatic method to compute a list of constraints that describe this model sufficiently well for Algorithms 2 and 3 to work correctly.
>
> The main applications we envision for our algorithms are as part of causal discovery methods. It is up to the designers of those methods what class of graphs they want to search, and so what graphs might be given as input to our Algorithm 2 & 3. The majority of existing causal discovery methods focuses on DAGs or MAGs, with very few works considering classes as general as BAPs. Thus for these applications, the restriction to BAPs is a very mild limitation.
>
>
> 2. [paper is not self-contained]
>
> We agree that the paper could be more self-contained by including more details from other papers where those details are relevant for this paper. We will clarify some definitions as suggested. By necessity, for some concepts a formal definition will not fit in this paper. E.g. we introduce the finite field $\mathbb{F}_p$ by writing "i.e. carrying out all computations modulo p." We will add a reference for readers who do want a full treatment. Similarly, for many aspects of algebraic geometry, HTC-identifiability, and nested Markov models / MArGs, we can only provide informal definitions in the paper.
>
> We will convert the definition of *model* $\mathcal{M}(G)$ (at the bottom of page 2) to a displayed equation, to make it easier to find. We believe this would answer your questions about the meaning of one model being included in another and the concept of closure.
>
> We plan to add an appendix where we can provide a thorough justification of the claims made in Examples 1 and 2, without cluttering the main text with technical details.
>
>
> 3. [practical demonstration of efficiency]
>
> We will add some specific information about running time to the paper, as this would indeed help to convince readers of the practical usefulness of these algorithms. When you refer to score-based algorithms, do you mean the use of these in Nowzohour et al. [2017]'s "empirical equivalence" to decide equivalence? In response to one of your other questions, it is indeed true that computing the score requires an iterative algorithm even in the linear case. And as another reviewer pointed out, the problem is nonconvex, leading to the possibility of false negatives due to ending up in a local maximum of the likelihood function.
>
> The only other existing algorithms for deciding algebraic equivalence are from algebraic geometry, e.g. based on Groebner bases (see Cox et al. [2015]). These algorithms may take hours or even days to complete, even for graphs with up to 5 nodes, whereas the algorithms presented here run in a matter of milliseconds.
>
>
> 4. [relevance for broader model classes?]
>
> We conjecture, but only proved partially (see Theorem 9), that algebraic equivalence is related to nested Markov equivalence. That would make the results in our paper relevant for discrete data, as well as for nonparametric models (i.e. we can answer your final question positively). Proving this relation is an important question for future work.
>
>
> We will reply to some of your remaining questions in a separate comment, due to the character limit on the rebuttal.

---

### Official Review · Reviewer_axpG · 2024-03-18

**Q2-1 Originality-Novelty:** 2
**Q2-2 Correctness-Technical Quality:** 3
**Q2-5 Clarity Of Writing:** 4

**Q1 Summary And Contributions:**

This paper studies the algebraic constraints (i.e., polynomial equality constraints on the covariance matrix) of linear structural equation models with bow-free acyclic path diagrams (BAPs).

Step by step the authors give three algorithms: 1) decides whether a graph imposes a specified algebraic constraint, 2) compares two graphs, and decides whether the algebraic model of the first is contained in that of the second, and 3) decides whether two graphs are algebraically equivalent.

Except for the algorithms, the authors also partially provide graphical criteria for the algebraic equivalence on BAPs. They argue that for BAPs the algebraic constraints provide the most fine-grained resolution achievable.

**Q2-3 Extent To Which Claims Are Supported By Evidence:**

3: Good: the main claims are supported by convincing evidence (in the form of adequate experimental evaluation, proofs, (pseudo-)code, references, assumptions).

**Q2-4 Reproducibility:**

2: Fair: key resources (e.g. proofs, code, data) are unavailable but key details (e.g. proof sketches, experimental setup) are sufficiently well-described for an expert to confidently reproduce the main results.

**Q3 Main Strengths:**

1. Significance. To the best of my knowledge, this is the first work that can determine the algebraic equivalence between a pair of linear SEM models (as of here, the bow-free acyclic path diagrams), as well as the algebraic constraints inclusions.

2. Clarity. The paper is clearly written. The concise descriptive texts with running examples make the main idea easy to follow. Logics are clear.

3. Comprehensiveness. The provided bounds on probability of error make the randomized algorithms more reliable.

**Q4 Main Weakness:**

1. Novelty is limited: Algorithm 1 is obvious, and the most primal Algorithm 2 (checks for the inclusion of algebraic models) is a merely a brutal extension on Algorithm 1: it first builds up the graphical ideal based on existing work, and then checks for the constraints one by one -- though authors use some tricks to "jointly" evaluate them, it is essentially still brutal enumeration. The later section on the relationship to distributional equivalence is also just result from existing work. Therefore, despite that it solves a new problem, there is not really anything totally new here, but more of an incremental on existing results/methods.

2. The randomized algorithms (sample parameters uniformly at random) can be way too inelegant and inefficient. Though the bounds on probability of error are given, intuitively a very large number of samplings (exponential scale) is needed to make the unfaithful cases less likely, especially when the graph scales up.

3. Though the algorithm to determine algebraic equivalence between two graphs is given, the corresponding graphical criteria, and furthermore, the characterization of the equivalence class, is still missing.

**Q5 Detailed Comments To The Authors:**

Here are some of my questions:

1. Regarding "We list some examples of algebraic constraints..", is there a way to *enumerate* all the graphical constraints associated with the algebraic constraints (i.e., is Van Ommen and Drton [2022] complete?) If not, is it possible to do so? Then, if it is possible, could the method given in this paper give any insights about it?

2. Regarding Section 4.1.1 "Graphical Conditions for Algebraic Equivalence", at least to me, the "Theorem 6 (Necessary condition)" and "Theorem 8 (Sufficient condition)" can be misleading -- at first I thought that authors give exactly the necessary and sufficient condition for two graphs' being algebraic equivalent. But it is actually not. Theorem 8 is ok, but Theorem 6 doesn't really give a descriptive/effective way to determine algebraic equivalence, as it 1) involves infinite enumeration on supergraphs, and it involves circular definition (for two supergraphs to be algebraic equivalent). I would suggest authors find some more appropriate names.

3. In Section 4.1, "Two graphs would fail to be distributionally equivalent *even if* a single Σ that is present in M(G) is missing in M(G′)." I didn't quite get the tone here. Is this a typo (i.e., "even" should be removed)?

4. It is argued that "the algebraic equivalence (i.e., distributional equivalence without ‘up to closure’) is the most fine-grained resolution *achievable*." Could the authors please elaborate more on what "achievable" means here? The inequality constraints do exist and can *achieve* for identification?

**Q9 Complying With Reviewing Instructions:**

Yes

---

> ### Author Rebuttal · Authors · 2024-04-03
>
> Thank you for your review of our paper, and your positive assessment of its significance as well as its clarity. You ask some interesting questions, which we discuss further below. First, let us try to address the points you mention as weaknesses of the paper.
>
> 1. "Novelty is limited [...]"
>
> We agree that Algorithm 1 is somewhat obvious (though the fact that the computations can be done modulo a large prime to achieve a low error bound is not obvious). While Algorithm 2 can be seen as a repeated application of Algorithm 1 with some improvements to keep the running time polynomial, what is definitely nontrivial is the proof that it is correct for BAPs. To see that this is nontrivial, note for the ADMG in Example 2, Algorithm 2 can have an error probability of 1! So an important contribution of this paper is to determine a class of graphs (namely BAPs) for which this algorithm does work.
>
> 2. [about the algorithms' efficiency]
>
> We disagree, and would say that the algorithms are very efficient. Because exact arithmetic is used, unfaithful cases only arise if the computation comes out as *exactly* 0 modulo p. We recommend to choose p large enough that a single run of the algorithm already gives a small probability of error. This is always possible: computers can handle p in the billions naturally (which is enough for graphs with up to 20 or so nodes), and can also handle arbitrarily large integers if needed (to deal with larger graphs).
>
> 3. [no graphical criteria or characterization of the equivalence class]
>
> We agree that a graphical characterization of algebraic equivalence is important future work. It is however outside the scope of this paper. We remark that graphical criteria for equivalence and efficient algorithms for checking equivalence are two separate problems, each with their own applications: cf. the case of Markov equivalence of MAGs, where a graphical characterization was given by Ali et al in 2009, but an $O(n^4)$ algorithm (the same complexity as our algorithm in practice) was not found until 2022 [Claassen and Bucur].
>
> In response to your other questions:
>
> 1. "Regarding "We list some examples of algebraic constraints..", is there a way to enumerate all the graphical constraints associated with the algebraic constraints (i.e., is Van Ommen and Drton [2022] complete?) If not, is it possible to do so? Then, if it is possible, could the method given in this paper give any insights about it?"
>
> This is a very interesting question. There are some ways in which the algorithm of Van Ommen and Drton [2022] is not complete: it works only on a subclass of ADMGs, and for a given input graph, it may not output enough constraints to describe the model as well as we might want (see Examples 1 and 2). Also, for a single algebraic constraints (i.e. a polynomial), there may be multiple graphical constraints representing it. Here the results of the present submission come in: with these algorithms, it can be checked efficiently if two BAPs are algebraically equivalent, even if their graphical constraints look different.
>
> 2. About "Theorem 6 (Necessary condition)" and "Theorem 8 (Sufficient condition)":
>
> Indeed, these are two separate conditions, one necessary and the other sufficient. The naming follows Nowzohour et al. [2017], and standard naming conventions for similar conditions. We will make sure the text of this section is clear on the fact that we do not have a condition that is simultaneously necessary and sufficient. We agree that given two BAPs, the condition of Theorem 6 would be a lot of work to verify completely, but for this purpose, Algorithm 3 should be used instead. We see these graphical conditions as fulfilling a different need; namely, they help with reasoning about algebraic equivalence classes.
>
> 3. "In Section 4.1, "Two graphs would fail to be distributionally equivalent even if a single $\Sigma$ that is present in M(G) is missing in M(G')." I didn't quite get the tone here. Is this a typo (i.e., "even" should be removed)?"
>
> Thanks for pointing that out, that should be "*if even* a single $\Sigma$ ...".
>
> 4. [the meaning of "achievable"]
>
> The statement about "most fine-grained resolution achievable" is made in the context of causal discovery for BAPs. By Theorem 5, BAP models don't impose inequality constraints, so that algebraic equivalence coincides with distributional equivalence up to closure. (The question says "*without* 'up to closure' apparently by mistake.) The only differences that may exist between two models that are distributionally equivalent up to closure, i.e. the only room to improve the resolution further, would be if some Sigma is in model 1 but not in model 2, but model 2 does contain other points arbitrarily close to Sigma. As argued before section 4.1.1, such models can't be distinguished using data alone. So the best we can hope for in causal discovery is finding the distributional-up-to-closure equivalence class.

---

### Official Review · Reviewer_xKnW · 2024-03-20

**Q2-1 Originality-Novelty:** 3
**Q2-2 Correctness-Technical Quality:** 3
**Q2-5 Clarity Of Writing:** 2

**Q1 Summary And Contributions:**

The paper considers the problem of deciding whether an ADMG and a BAP are algebraically equivalent. To do so they first consider the relationship between the algebraic ideal generated by the equalities that hold in all covariance matrices compatible with a graph and the graphical ideal. They establish that for BAPs this difference is sufficiently small that checking whether the two graphs have the same graphical ideal suffices to verify that they have the same algebraic ideal, i.e., are algebraically equivalent. Building on this result the authors propose i) a Monte Carlo algorithm that is likely to correctly verify whether a graphical model satisfies some constraint and ii) a Monte Carlo algorithm that is likely to correctly verify whether an ADMG is algebraically equivalent to BAP with high probability. Finally the authors discuss the relationship of algebraic equivalence to other graphical equivalence notions and under what assumptions their algorithms can be used to verify these alternatives equivalences as well.

**Q2-3 Extent To Which Claims Are Supported By Evidence:**

2: Fair: the main claims are somewhat supported by evidence (but the experimental evaluation may be weak, or does not match entirely with the claims, important baselines may be missing, proofs contain important ideas but lack rigor, algorithmic details are only discussed superficially, references are imprecise, assumptions are not sufficiently motivated or explicated, etc.).

**Q2-4 Reproducibility:**

3: Good: key resources (e.g. proofs, code, data) are available and key details (e.g. proofs, experimental setup) are sufficiently well-described for competent researchers to confidently reproduce the main results.

**Q3 Main Strengths:**

Graph equivalence is a notion central to causal discovery. Improving our understanding of the subtle non-CI constraints that arise in models with hidden variables is also a very interesting mathematical problem. As a result I find the topic of the paper theoretically very interesting, although one may argue that the topic is of a limited practical relevance. While the paper mostly combines existing results rather than proving completely new ones, it does so in innovative ways, so I nonetheless find its technical contribution meaningful.

**Q4 Main Weakness:**

Since the area the paper covers, the algebraic statistics of linear SEMs, is very dense the authors necessarily had to at some point limit the depth at which they introduce certain concepts, e.g., the graphical ideal or HTC-identifiability. As a non-expert in the area I am uncertain whether this is unavoidable but I do believe that it does limit the comprehensibility of the paper at times. I in particular struggled in Section 3.2 where I could not follow how exactly Theorem 2 exactly evaluates whether the constraints in G hold in G' without explicitly doing so. For example, I don't even understand why it was necessary to compute new parameters for G' (sentence in the middle of the paragraph on the left-hand side of page 5). I make some further comment regarding the presentation in the detailed comments.

**Q5 Detailed Comments To The Authors:**

- Sec. 1.1 bottom paragraph: at this point I don't think it is yet clear what it means for one model to contain another.
- Sec 2 first paragraph: I would not recommend connecting the presence of an edge with the notion of cause since in general ancestors are though of as causes (and not just parents). I understand that the concept of a direct cause is also problematic, however.
- Page 3, bottom right. "A graph is HTC-identifiable ...." I believe there is an if missing. In the following sentence there is a bracket missing.
- I don't understand the last sentence in Example 2.
- Sec 3.1 first paragraph: can you make "most" more precise?
- Proof of Theorem 2: can there be cases where a polynomial is 0 on all elements of F_p for a small p?
- Section 3.2. in general: as said above I have a hard time following the explanation of the algorithm.

**Q9 Complying With Reviewing Instructions:**

Yes

---

> ### Author Rebuttal · Authors · 2024-04-03
>
> Thank you for your comments, which will help to improve the presentation of the paper. We appreciate your question about the explanation of Algorithm 2 in Section 3.2, as indeed we want the intuition of the algorithm to be clear from the main paper. We will clarify the following in the revised version of the paper: (The sentence in italics below addresses your question of *why* the algorithm computes these parameters.)
>
> The intuition behind the graphical constraints of Van Ommen and Drton [2022] is that first, $\Lambda'$ is computed using [Foygel et al., 2012]. This algorithm will always assign 0's to elements of $\Lambda'$ that should be 0, i.e. those that don't correspond to directed edges in $G'$. Next, $\Omega'$ is computed as $(I - \Lambda')^T \Sigma (I - \Lambda')$. This computation does not check where in $\Omega'$ it places nonzeros. If $\Sigma$ was in the model of $G'$, then $\Omega'$ will have its nonzeros only on the diagonal and in places where $G'$ has bidirected edges. But if $\Sigma$ was not in the model of $G'$, $\Omega'$ will typically have nonzeros in certain other places as well. *Computing the values of these elements of $\Omega'$ amounts to evaluating each of the graphical constraints* -- the only difference being that the graphical constraints are polynomials in $\Sigma$, while computing $\Lambda'$ (and thus $\Omega'$) from $\Sigma$ also requires divisions. Algorithm 2 avoids these divisions by computing polynomial multiples of $\Lambda'$ and $\Omega'$ instead, thereby mimicing the computation of Van Ommen and Drton [2022] exactly. Thus Algorithm 2 computes the matrix $\tilde{\Omega}'$, whose entries are multiples of $\Omega'$. Because $I - \Lambda'$ plays a more central role in this computation that $\Lambda'$, it is convenient in Algorithm 2 to work with $\tilde{\Lambda}'$, which equals $I - \Lambda'$ except that each row is multiplied by some polynomial.
>
> You raise some other questions in the detailed comments. Our responses to these are below:
>
> - "I don't understand the last sentence in Example 2."
>
> (assuming you mean the sentence starting "In other words...") The desired situation is that a positive definite $\Sigma$ satisfies the graphical constraints iff $\Sigma$ is in the algebraic model. But this sentence states that for this example, the set of points that satisfy the graphical constraints is much larger, and also contains the set {$\Sigma$ : $\sigma_{ac} = \sigma_{ad} = 0$}.
>
> - "Sec 3.1 first paragraph: can you make "most" more precise?"
>
> Yes, there are two ways to make this statement precise. One is in Theorem 1, which uses the concept of dimension from algebraic geometry. The other one is using the Schwartz-Zippel lemma, which provides a bound to the number of zeros in certain finite sets; this is the version that is used in the proofs of Theorems 2 and 3. We will mention these two precise statements at this point in the paper.
>
> - "Proof of Theorem 2: can there be cases where a polynomial is 0 on all elements of F_p for a small p?"
>
> Yes, this is possible (at least for general polynomials; not sure if it can happen for the polynomials that appear in the algorithms). In such a case, the upper bound on the probability of error will be $\geq 1$, i.e. vacuous. The bound will become useful again for a sufficiently large p.

---

### Official Review · Reviewer_owGh · 2024-03-23

**Q2-1 Originality-Novelty:** 3
**Q2-2 Correctness-Technical Quality:** 3
**Q2-5 Clarity Of Writing:** 3

**Q1 Summary And Contributions:**

This paper discusses the notion of algebraic equivalence for BAPs. They provide 3 algorithms which: checks if an ADMG satisfies an algebraic constraint, checks if the algebraic constraints entailed by an ADMG are a subset of those entailed by a BAP, and checks if two BAPs are algebraically equivalent, respectively. These algorithms are random and can give incorrect output with small probability (which can be reduced with more computation). These algorithms are much more tractable than existing methods. Lastly, the authors argue for the use of algebraic equivalence classes in causal discovery for BAPs.

**Q2-3 Extent To Which Claims Are Supported By Evidence:**

3: Good: the main claims are supported by convincing evidence (in the form of adequate experimental evaluation, proofs, (pseudo-)code, references, assumptions).

**Q2-4 Reproducibility:**

3: Good: key resources (e.g. proofs, code, data) are available and key details (e.g. proofs, experimental setup) are sufficiently well-described for competent researchers to confidently reproduce the main results.

**Q3 Main Strengths:**

The paper presents algorithms for equivalence and containment of BAP models wrt algebraic constraints that greatly improve over current methods.

**Q4 Main Weakness:**

This work relies heavily on existing work and it hard to follow without being familiar with the work on HTC identifiability.

**Q5 Detailed Comments To The Authors:**

In regard to how Nowzohour et al. test empirical equivalence, you could also note that the space they are trying to compute the MLE within is not convex; see Drton and Richardson 2004.

For those unfamiliar with BAPs, you could describe them as simple ADMGs (an ADMG without multiple edges.)

The definition of half-trek and htr are not as descriptive as they could be. For instance, the direction of the "directed path" is ambiguous (is "<-> <- ... <-" a half-trek?). It is also not obvious from the definition that the siblings of a vertex are not in htr.

Typo: "A graph is HTC-identifiable sets Yv exist ..."   =>   "A graph is HTC-identifiable if sets Yv exist ..."

In the "Vanishing partial correlation" example of an algebraic constraints, should the semicolon in the subscript of Sigma be a comma to match the notation used elsewhere in the paper?

You say: "In the former case, we can be sure of the correctness of the answer; in the latter case, the probability of error has been reduced to q^k." Is it the case that the repeated runs of the randomized algorithm are independent?

At the end of section 3.2 when you are discussing the computation time, it would be nice to note anecdotally how long (wall time) these computations took (for n=5 and n=25 w/ repeated runs.) I appreciate that your algorithm is many times more efficient than what was previously available, but I think making a note of this nature would sate many readers' curiosity.

I think there may be a typo in this sentence: "It turns out that the same criteria are necessary resp. sufficient also for algebraic equivalence of BAPs."

After Theorem 7 and 8, I recommend noting these theorems imply that there may be BAPs that are distributionally/algebraically equivalent that have the same skeleton and v-structure but different collider triples.

You say: "Markov equivalence is coarser than algebraic equivalence, as it considers only (conditional) independence constraints, which form a strict subset of algebraic constraints." Right below this you say algebraic equivalence is the same as Markov equivalence for MAGs so the use of the word "strict" here seems incorrect.

You say: "Algorithms 2 and 3 can also be used to decide inclusion and equivalence of nested Markov models, by first applying the maximal arid projection to the input graphs." Are there not graphical criteria wrt MArGs that do this? If not, I would explicitly say so as it is a bug selling point for your work.

**Q9 Complying With Reviewing Instructions:**

Yes

---

> ### Author Rebuttal · Authors · 2024-04-03
>
> We'd like to thank the reviewer for their useful comments about related literature, and other insightful suggestions that will improve the presentation of this paper. In particular, these comments, as well as those from the other reviewers, will help to make the paper more self-contained, so that the core ideas are clear even if details of some of the cited work are unfamiliar. We are happy to hear you enjoyed the paper!
>
> Below are our answers to the questions you asked:
>
> - "You say: [...] Is it the case that the repeated runs of the randomized algorithm are independent?"
>
> Repeated runs of the algorithm are indeed independent (under standard assumptions on the pseudorandom number generator): the randomness comes from sampling the parameters in the first step of algorithms 1 and 2, and so a new run will give an independent sample from the same distribution.
>
> - 'I think there may be a typo in this sentence: "It turns out that the same criteria are necessary resp. sufficient also for algebraic equivalence of BAPs."'
>
> Do you mean that "respectively" should be written in full?
>
> - 'You say: "Markov equivalence is coarser than algebraic equivalence, as it considers only (conditional) independence constraints, which form a strict subset of algebraic constraints." Right below this you say algebraic equivalence is the same as Markov equivalence for MAGs so the use of the word "strict" here seems incorrect.'
>
> Indeed, the word "strict" applies to the set of all algebraic constraints, not if we restrict attention to constraints that arise in MAGs. We will clarify this.
>
> - 'You say: "Algorithms 2 and 3 can also be used to decide inclusion and equivalence of nested Markov models, by first applying the maximal arid projection to the input graphs." Are there not graphical criteria wrt MArGs that do this? If not, I would explicitly say so as it is a bug selling point for your work.'
>
> To our knowledge, no such criteria exist for nested Markov equivalence of MArG. Because Theorem 9 only proves one direction and the other direction is now only a conjecture, we did not put this aspect of our work in the spotlight. But we agree that right now it is too hidden away.

---

### Meta-Review · Area_Chair_U1Cu · 2024-04-17

The paper's main results have the potential to improve the information gained by applying causal discovery algorithms when unmeasured confounders are present. Initial reviews were mixed, with one strong accept, a weak accept, and three reviewers leaning towards rejection. One of the reviewers leaning towards rejection revised their decision to leaning towards acceptance after the discussion. Some of the main reasons brought up for rejection was the vast amount of background knowledge assumed of the reader regarding HTC-identifiability and other graph identification concepts along with algebraic geometry. The authors have proposed solutions to rectify this problem like adding more information about core concepts to the appendices.

More detailed thoughts about positive aspects of the paper are below.

- Equivalence criteria for causal models with unmeasured confounders is a fundamental obstacle to the development of new causal discovery algorithms in settings that go beyond causally sufficient systems

- Causal models with unmeasured confounders exhibit more complex constraints than ordinary conditional independences, but very little is known about deciding equivalence between two causal models with respect to these more complex constraints, e.g., Verma constraints

- The authors make progress on this problem by proposing Monte Carlo algorithms that decide algebraic equivalence correctly with high probability -- this is a novel and fascinating approach to the equivalence problem. It is interesting to think about whether this probabilistic approach is "good enough" for downstream causal inference tasks and how one would encode the uncertainty in this.

- The authors also provide interesting examples and connections of their method, which is designed for linear SEMs, to other coarser notions of equivalence that are non-parametric e.g., nested Markov equivalence

Below are some remarks on points the paper could improve on.

- The authors should take to heart the comments provided by the reviewers regarding the readability of the paper and incorporate the promised changes in any revisions to the paper.

- In addition, I have a clarifying question about Algorithm 1 --- how does the algorithm ensure that sampling from $F_p$ produces a positive definite covariance matrix? I didn't find a place where this was explicitly addressed.

- I also suggest the authors add a paragraph or so about the application of their method to causal discovery --- for example, in the discussion about MArGs, it might be helpful to connect the work to discovery algorithms, such as https://arxiv.org/pdf/2010.06978.pdf, that operate on arid (but not necessarily maximal arid) graphs. Another class of algorithms worth considering (and seemingly related to this work) are the ones that use SAT solvers (e.g., https://arxiv.org/pdf/1309.6836.pdf and other follow-up work) to check satisfiability of a list of constraints --- these also typically go beyond regular Markov equivalence.